# DialSim: A Dialogue Simulator for Evaluating Long-Term Multi-Party Dialogue Understanding of Conversational Agents

## Abstract

Recent advancements in Large Language Models (LLMs) have significantly enhanced conversational agents, making them applicable to various fields (*e.g.*, education, entertainment). Despite their progress, the evaluation of the agents often overlooks the complexities of real-world conversations, such as multi-party dialogues and extended contextual dependencies. To bridge this gap, we introduce DialSim, a dialogue simulation-based evaluation framework. In DialSim, an agent assumes the role of a character in a scripted conversation and is evaluated on their ability to answer spontaneous questions using only the dialogue history, while recognizing when they lack sufficient information. To support this framework, we introduce LongDialQA, a new QA dataset constructed from long-running TV shows, comprising over 1,300 dialogue sessions, each paired with more than 1,000 carefully curated questions, totaling over 352,000 tokens. To minimize reliance on prior knowledge, all character names are anonymized or swapped. Our evaluation of state-of-the-art LLM-based conversational agents using DialSim reveals that even models with large context windows or RAG capabilities struggle to maintain accurate comprehension over long-term, multi-party interactions—underscoring the need for more realistic and challenging benchmarks in conversational AI.

## 1 Introduction

Recent advancements in Large Language Models (LLMs) have significantly enhanced the capabilities of conversational agents. These agents are now applied across various domains, including entertainment (Zhou et al., 2023; Chen et al., 2024) and education (Ait Baha et al., 2023; Waisberg et al., 2024), providing more natural interactions that enhance user satisfaction. As these agents become increasingly integrated into real-world applications, it is essential to evaluate their performance in realistic conversational settings.

Real-world conversations present a range of challenges that make them difficult for conversational agents to handle effectively. They are often (1) **long-term**, requiring agents to retain information over extended interactions and perform (2) **multi-hop reasoning**, as they must connect details spread across multiple turns or even sessions to understand the context and respond appropriately. These conversations are also frequently (3) **multi-party**, involving several participants whose inputs must be interpreted in relation to one another. Moreover, real-world dialogue often involves ambiguity or incomplete information, so agents need to (4) **handle uncertainty gracefully**, including recognizing when they lack sufficient knowledge to provide a reliable answer.

However, existing evaluation approaches are insufficient to capture these realistic scenarios. Traditional methods primarily assess agent response quality in terms of fluency, naturalness, and alignment with a given instruction (Roller et al., 2021; Shuster et al., 2022; Lee et al., 2023; Kim et al., 2024; Chiang et al., 2024). These evaluations are typically based on single-turn instructions or brief dialogues, and thus fail to account for performance in long-term, multi-party conversations or in scenarios involving uncertainty. More recently, several studies have sought to address these limitations by proposing question-answering (QA) benchmarks based on long-term dialogues to evaluate conversational capabilities. However, these datasets are limited in that they do not feature multi-party dialogue, and often involve relatively short conversations, typically under 10,000 tokens (Maharana

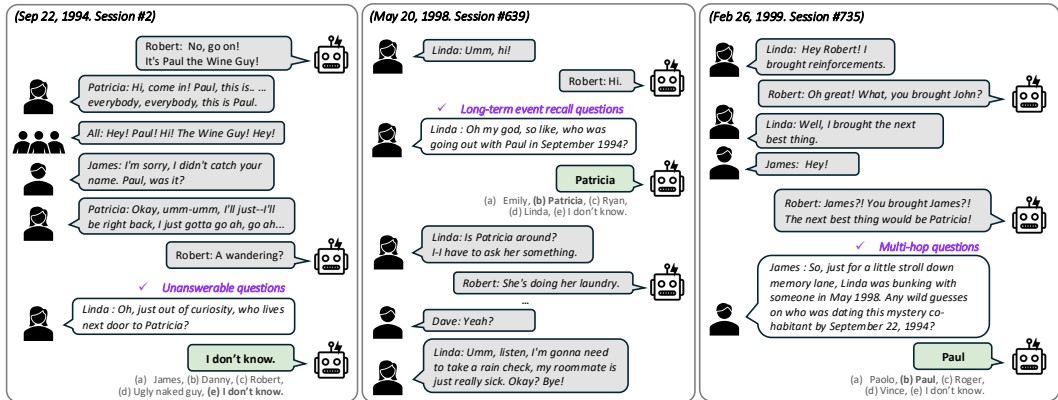

Figure 1: An overall process of DialSim. Gray bubbles represent scripted utterances, and white speech bubbles indicate spontaneous questions asked during the simulation. Colored speech bubbles indicate the agent's responses to the questions. (Left) An unanswerable question. (Center) A long-term event recall question. (Right) A multi-hop question that requires understanding past sessions (*i.e.*, the Left and Center boxes). The dialogue and questions are based on the Friends script, with character names anonymized (*e.g.*, Ross → Robert). The question is asked in the format chosen by the user, either in a multiple-choice format or as an open-ended question.

et al., 2024), or focus on AI-user interactions where consecutive dialogue sessions are independent, lacking continuity across sessions (Wu et al., 2025).

To address these limitations, we propose **DialSim**, a simulation-based framework for evaluating the dialogue understanding of conversational agents. In DialSim, an agent is assigned the role of a specific character in a predefined dialogue and participates in the extended multi-party dialogue as it progresses (see Figure 1). During the interaction, the agent is randomly asked questions by other participants at unpredictable times. The agent must respond appropriately based solely on the dialogue history, and acknowledge when it lacks sufficient information to answer confidently. This simulation-based evaluation method closely mirrors real-world conversations, enabling a rigorous assessment of an agent's dialogue comprehension in unpredictable settings.

To implement DialSim, a dialogue script and corresponding QA pairs are required. For this purpose, we created **LongDialQA**, a new QA dataset derived from long-term multi-party dialogues. It comprises dialogues from popular TV shows (*i.e.*, Friends, The Big Bang Theory, and The Office), covering 1,300 sessions over five years, totaling around 352,000 tokens. Each session includes more than 1,000 questions curated through two approaches: refining questions from a fan quiz website and generating complex questions using the temporal knowledge graphs extracted from the dialogue script. At each stage of question generation, GPT-4 (OpenAI, 2023) assisted in refining the questions and extracting knowledge graphs, allowing the authors to thoroughly review and ensure quality. After constructing the QA pairs, we anonymized the names of main characters in both the dialogues and questions by assigning generic names (*e.g.*, Joey → John), thereby mitigating the influence of any prior knowledge that LLMs may have about the shows (see Appendix A). We also provide a more adversarial version of LongDialQA in which character names are swapped with each other (*e.g.*, Joey ↔ Monica). These modifications ensure that agents must rely solely on the dialogue context, rather than any pre-trained knowledge of the TV shows.

We then built a range of conversational agents using recent LLMs, leveraging either their extended context capabilities or RAG techniques (Lewis et al., 2020), and evaluated them using DialSim. As a result, none of the agents scored above 60%, and even those with extended context windows (128K to 1M tokens) struggled to understand dialogue histories spanning 352K tokens. These results highlight the significant limitations that current conversational agents still face in accurately understanding and tracking long-term multi-party dialogues.

## 2 RELATED WORKS

**Conversational Agents Evaluation** Early evaluation methods for conversational agents often relied on reference-based metrics (e.g., BLEU (Papineni et al., 2002), ROUGE (Lin, 2004), ME-

Table 1: Comparison of LongDialQA with long-term dialogue datasets. Dialogue Length is measured in tokens.

| Dataset | Dialogue Length | Avg. # of Speakers | QA pairs | Session Continuity |
|---|---|---|---|---|
| MSC (train) | 1.2k | 2.0 | X | O |
| MSC (valid + test) | 1.6k | 2.0 | X | O |
| Conversation Chronicles | 1.0k | 2.0 | X | O |
| LoCoMo | 9.2k | 2.0 | O | O |
| LongMemEval | 115k, 1.5M | 2.0 | O | X |
| **LongDialQA (ours)** | **352k** | **3.4** | **O** | **O** |

TEOR (Banerjee & Lavie, 2005)), which compare model outputs to gold dialogue references but often show weak correlation with human judgment (Liu et al., 2016). In contrast, human evaluation—where human annotators assess coherence, factual correctness, consistency, and engagingness of the generated responses—provides reliable assessments (Zhang et al., 2020; Roller et al., 2021; Shuster et al., 2022; Lee et al., 2023), but it is costly and time-consuming.

With the advent of LLMs, new evaluation approaches have emerged. These include having LLMs evaluate utterances directly (Li et al., 2023; Kim et al., 2024) or employing platforms (*e.g.*, *Chatbot Arena* (Chiang et al., 2024)) where humans rank responses from different agents. Despite these advances, existing methods are still limited to qualitative assessments of utterances and fail to capture real-world conversational scenarios (*e.g.*, long-term multi-party dialogue).

**Long-Term Dialogue Datasets** Multi Session Chat (Xu et al., 2022) introduced a dataset containing up to five sessions per dialogue, marking a step forward in modeling extended interactions. However, generating longer and coherent dialogues through crowdsourcing remained a challenge. To address this, Conversation Chronicles (Jang et al., 2023) leveraged LLMs to generate more extended and coherent dialogue sessions. More recently, LoCoMo (Maharana et al., 2024) was proposed to evaluate an agent's dialogue comprehension abilities through tasks such as event summarization. In addition, LongMemEval (Wu et al., 2025) was introduced as a QA dataset to evaluate whether an agent can understand long-term interactions—up to 1.5M tokens—between an AI and a user. While these datasets contribute valuable resources for long-term dialogue research, they have several limitations. All are limited to two-party interactions, and most involve relatively short dialogues, typically under 10k tokens. Although LongMemEval features much longer dialogues, it lacks continuity across sessions, as each interaction is treated independently without sustained contextual linkage (Wu et al., 2025). Table 1 provides a detailed comparison between LongDialQA and other existing long-term dialogue datasets.

## 3 LONGDIALQA

To implement DialSim, we first developed LongDialQA, a QA dataset derived from long-term multi-party dialogues.

### 3.1 DATA CONSTRUCTION

LongDialQA was developed using scripts from five consecutive seasons of popular TV shows (*i.e.*, Friends, The Big Bang Theory, and The Office[1]). These scripts were first preprocessed to serve as dialogue data. Next, questions were generated for each script, drawing from fan quizzes and a temporal knowledge graph (TKG). Each question was then paired with the correct answer and multiple distractors. Finally, character style transfer was applied to refine the questions, resulting in the final pool of questions for each session.

#### 3.1.1 SCRIPT PREPROCESSING

The script we used includes 5 consecutive seasons per TV show, with each season containing approximately 20 episodes. Each episode is composed of multiple scenes (*i.e.*, session). Each script

---

[1]The scripts were downloaded from the website Kaggle (https://www.kaggle.com/).

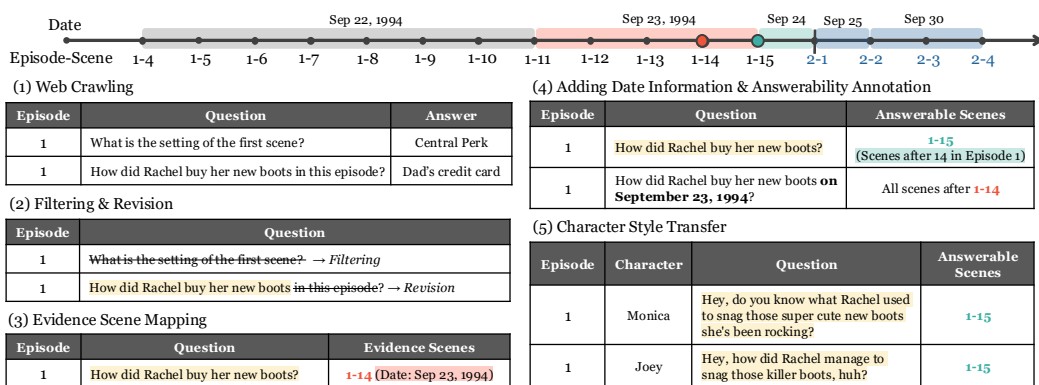

Figure 2: The overall process of question generation based on fan quizzes. First, we crawled fan quizzes from the web (1). Next, we applied filtering and revision processes to the crawled data (2). After that, we identified evidence scenes that could provide answers to the questions (3). From this, we created secondary versions of the questions by adding dates to each. We then mapped each question to the scenes by determining whether it is answerable in that scene or not (4). Finally, we applied character style transfer to make the questions more natural (5).

includes not only utterances but also descriptions of characters' actions and scenes, as well as meta-data unrelated to the plot (*e.g.*, names of writers and directors). We manually filtered out all irrelevant parts to create $Script_{pre}$, which contains only the conversations between characters. Additionally, since some of our questions involve time conditions (*e.g.*, "Which friend wasn't allowed to drive Monica's Porsche in October 1994?"), we manually assigned a date to each scene in $Script_{pre}$ to provide time information to the agent. These dates were determined based on the contents of the conversations and the air dates of the episodes. The specific rules for date assignments are detailed in Appendix B. We then selected scenes involving the main character (*i.e.*, Friends: Ross, The Big Bang Theory: Sheldon, The Office: Michael[2]) from $Script_{pre}$ and sequentially numbered them as sessions $S_i$. This process resulted in the final dialogue $\mathcal{D} = \{\mathcal{S}_1, \mathcal{S}_2, ..., \mathcal{S}_N\}$.

### 3.1.2 FAN QUIZ-BASED QUESTION GENERATION

We utilized a fan quiz website FunTrivia[3] to generate our questions. Fan quizzes cover a range of difficulty levels and focus on major events from each episode, making them promising for evaluating dialogue comprehension. Figure 2 illustrates our process for generating questions using fan quizzes. We began by extracting episode-specific quizzes from the site. Since these quizzes were created by dedicated fans, many required knowledge unrelated to the dialogue itself (*e.g.*, "What is the name of the actor who played the clerk?"). To filter out these questions, we first selected quizzes that could be answered by referencing $Script_{pre}$ using GPT-4 (OpenAI, 2023).[4] Additionally, GPT-4 annotated the scenes that served as evidence for each question. The authors verified these annotations to ensure accurate filtering and scene-mapping.

We then annotated the answerability of each question, i.e., whether it is possible for the main character to know the answer in the corresponding scene. For example, in Friends, if the evidence for a question was in scene 14, Ross would not know the answer if he was absent from that scene. Even if he were present in scene 14, he couldn't answer the question if it had been asked in scene 1. However, if Ross appeared in scene 14 and the question was then asked in scene 15, he would know the answer. Using this principle, we determined whether each question is answerable. Additionally, to create questions that require long-term memory, new questions were generated by adding the date information of each scene to the questions (*e.g.*, "How did Rachel buy her new boots on September 22, 1994?"). Detailed question generation processes are provided in Appendix C.

---

[2]The characters with the most lines in each script were selected.

[3]https://www.funtrivia.com/

[4]Fan quizzes exist for each episode, so we annotated them based on $Script_{pre}$ and then matched them to the sessions of $\mathcal{D}$. Questions about scenes without the main character are unanswerable, enabling us to design rigorous tests.

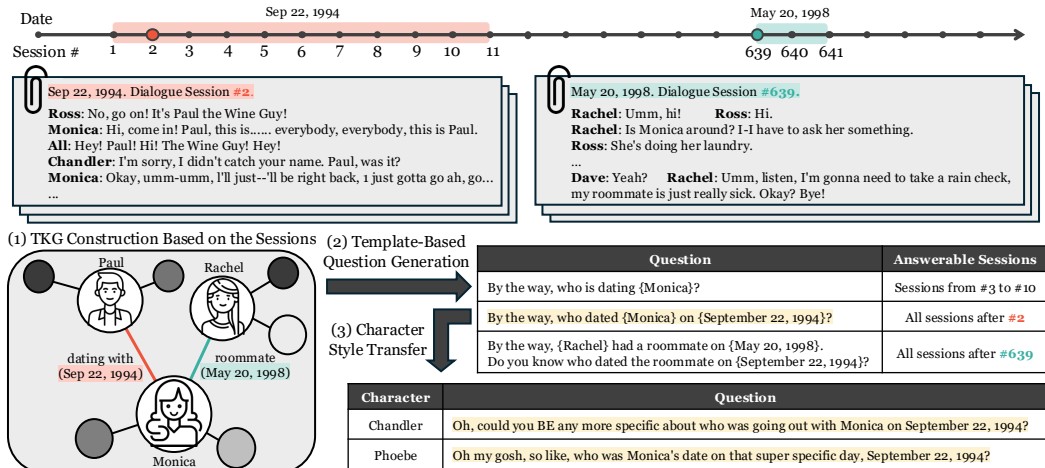

Figure 3: The overall process of question generation based on the temporal knowledge graph. We first extracted quadruples and constructed a temporal knowledge graph (1). Then, we generated questions based on this and mapped each question to the sessions by determining whether it was answerable in that session or not, similar to fan quiz-based questions (2). Character style transfer was performed afterwards (3).

### 3.1.3 TEMPORAL KNOWLEDGE GRAPH-BASED QUESTION GENERATION

Fan quizzes are useful for generating our questions, but since they are episode-specific and user-generated, the questions don't span multiple episodes and their numbers are limited (∼1k). To address this, we constructed a knowledge graph for each session and used it to generate questions. Initially, we used GPT-4 to extract triples (*i.e.*, [head, relation, tail]) from each session $S_i$ in $\mathcal{D}$. These triples were then refined by the authors. We employed 32 relations (*e.g.*, girlfriend) derived from DialogRE (Yu et al., 2020), a high-quality dataset where human annotators manually extracted relations from Friends scripts, classifying relationships between characters into 37 categories. We adapted and modified these relations for our purpose. More details about the relations are provided in Appendix D.1. Finally, we combined the triples from each session with their respective dates to create a temporal knowledge graph (TKG) composed of quadruples (*i.e.*, [head, relation, tail, date]).

Using the TKG, we created questions that the main character could either answer or not for each session. We generated these questions by extracting one (*i.e.*, one-hop) or two (*i.e.*, two-hop) quadruples from the TKG. The form and answer of the question may change depending on the time it is asked, even if the same quadruple is used. For instance, if we select [Rachel, boyfriend, Ross, 1994-08-08] and ask the question in 1996, it would be: "Who was Rachel's boyfriend on August 8th, 1994?" If asked on August 8th, 1994, the question would be: "Who is Rachel's boyfriend?" In both cases, the answer is Ross. Conversely, if we inquire about Rachel's boyfriend in 1992, when no information is available, the correct answer would be: "I don't know." In this manner, we manually verified the answer of each question. We applied the same principle to create more complex two-hop questions (*e.g.*, "Rachel had a roommate on August 8th, 1994. Who is the boyfriend of the roommate now?"). The overall process of generating questions using TKG is illustrated in Figure 3. Examples of question templates and generated questions are provided in Appendix D.2.

### 3.1.4 ANSWER CHOICES GENERATION

To create multiple-choice questions, we carefully crafted a set of answer choices for each question. First, for all questions, we included a choice "(E) I don't know.", which agents must choose if the questions are unanswerable. For questions sourced from fan quizzes, the four answer choices were taken from the original quiz. The correct answers for these questions were the same as the original quiz, while the unanswerable questions were fixed to (E).

For TKG-based questions, the incorrect choices were derived from the tails of other quadruples that shared the same relation as the original quadruple. For example, for the question "Who is Rachel's

Table 2: Statistics of LongDialQA. † indicates the average number of questions per session.

|  | Friends | The Big Bang Theory | The Office |
|---|---|---|---|
| Dialogue Length (tokens) | 335,439 | 367,636 | 352,914 |
| Number of Dialogue Sessions | 788 | 805 | 2,347 |
| Fan Quiz Questions[†] | 192.9 | 26.7 | 42.7 |
| TKG Questions[†] | 1173.2 | 1280.1 | 455.1 |
| Total Question Candidates[†] | 1366.1 | 1306.8 | 497.9 |
| ↪Answerable Questions[†] | 1215.0 | 1239.7 | 410.9 |
| ↪Unanswerable Questions[†] | 151.1 | 67.2 | 86.9 |
| **Approximate Number of Possible Tests** | $1366.1^{788}$ | $1306.8^{805}$ | $497.9^{2347}$ |

boyfriend?", we extracted quadruples from the whole TKG where the relation is "boyfriend" and randomly selected three tails to form the incorrect choices. Additionally, to create a more adversarial test, if Rachel has a boyfriend in the past or future, we prioritized including these in the incorrect choices. In this case, for answerable questions (*i.e.*, past or present), the correct answer is the tail of the original quadruple, while for unanswerable questions (*i.e.*, future), the correct answer is (E).

### 3.1.5 QUESTION STYLE TRANSFER

In LongDialQA, questions are rephrased to reflect each character's unique tone, creating the impression that the characters themselves are asking the questions (*e.g.*, Generic style: "How did Rachel buy her new boots?"→ Style of Joey Tribbiani from Friends: "Hey, how did Rachel manage to snag those killer boots, huh?"). This transformation is powered by GPT-4, and subsamples are reviewed by the authors to ensure that the original intent was preserved. More examples of style-transferred questions for each character are in Appendix E.

### 3.1.6 CHARACTER NAME ANONYMIZATION

We replaced original character names with generic placeholders (*e.g.*, Joey → John), ensuring that agents must rely on contextual reasoning rather than prior knowledge of the TV shows. In addition to this anonymization, we created a more adversarial version by swapping the names of characters (*e.g.*, Joey ↔ Monica). This method can further confuse agents by inducing dialogues that contradict the characteristics they may have memorized about the original characters.

### 3.2 STATISTICS

Table 2 presents the statistics of LongDialQA, highlighting a significant disparity between the number of answerable and unanswerable questions. When conducting experiments using DialSim in the form of multiple-choice questions, to ensure a balanced distribution of correct answers during the simulation, 20% of the questions were intentionally designed to be unanswerable, with each question providing five possible choices.

## 4 DIALSIM

Building on LongDialQA, DialSim features an agent taking on the role of a main character in a dialogue. Throughout the simulation, the agent is asked questions by other characters that must be answered accurately.

Algorithm 1 outlines the simulation process of DialSim, designed to emulate a conversation. In this simulator, each participant's utterance (including the agent's) occurs, and the agent should update its memory.[5] During the simulation, other characters ask questions (selected from LongDialQA) to the agent (Line 8-10), except in sessions where the agent is the only one talking (Line 5-6). The timing

---

[5]The memory can be incrementally updated in various ways (*e.g.*, by storing each utterance separately or by summarizing the session up to the current utterance). A detailed discussion of these methods is provided in § 5.2.

---

**Algorithm 1:** DialSim

---

**Input:** $\mathcal{D} = \{\mathcal{S}_i\}_{i=1}^{N}$, Agent
**Output:** $C/T$ (CorrectAnswers / TotalQuestions)

1: $C \leftarrow 0$ // CorrectAnswers
2: $T \leftarrow 0$ // TotalQuestions
3: $\mathcal{M}_{1,0} \leftarrow \varnothing$;
4: **for** $n \leftarrow 1$ **to** $N$ **do**
5:      **if** $|Characters(\mathcal{S}_n)| < 2$ **then**
6:          **continue**
7:      **else**
8:          $u_{n,m} \leftarrow SelectQuestionTiming(\mathcal{S}_n)$;
9:          $c \leftarrow RandCharInThreeTurns(u_{n,m})$;
10:         $(q_{n,m,c}, a_{true}) \leftarrow RandomQnA(n, m, c)$;
11:         $T \leftarrow T + 1$;
12:         **for** $k \leftarrow 1$ **to** $|\mathcal{S}_n|$ **do**
13:            $\mathcal{M}_{n,k} \leftarrow UpdateMemory(\mathcal{M}_{n,k-1}, u_{n,k}, d_n)$;
14:            **if** $k = m$ **then**
15:              $a_{n,m} \leftarrow AgentAns(\mathcal{M}_{n,m}, q_{n,m,c}, d_n)$;
16:              **if** $a_{n,m} = a_{true}$ **then**
17:                 $C \leftarrow C + 1$;
18:            $\mathcal{M}_{n+1,0} \leftarrow \mathcal{M}_{n,k}$;

19: **return** $C/T$

---

to ask a question is chosen randomly within the session (Line 8), and the speaker who asks the question is also chosen randomly. However, to make the simulation realistic, it is crucial to ensure that the chosen speaker is still present and hasn't left the session. We achieved this by randomly choosing from characters who were present within three turns of the agent's last utterance (Line 9). Then, a question is randomly selected and asked in the style of the corresponding speaker (Line 10). The agent then must respond to the question using its memory (Line 15). The prompt for the response is created by combining the question with the dialogue history stored in the memory. The prompt we used is provided in Appendix F.

## 5 EXPERIMENTS

### 5.1 EXPERIMENTAL SETTING

We provide the option to choose between **multiple-choice and open-ended question formats**. In our experiments, we used the multiple-choice format to efficiently and accurately assess the agent's dialogue understanding capabilities. Details about the question formats and the open-ended questions can be found in Appendix G. The temperature for the LLMs was set to 0.2, and the top-p value was set to 0.1. All experiments were conducted using NVIDIA RTX A6000 GPUs and an AMD EPYC 7702 64-Core Processor.

### 5.2 CONVERSATIONAL AGENTS

We experimented with LLM-based conversational agents capable of handling long-term dialogue, focusing on two memory management approaches. The first method, namely Base LLM, simply prepends as many of the most recent utterances as the model's context length allows. The second method, namely RAG-based, employs a retriever to search for relevant dialogue history from the agent's memory (external storage) and includes it in the prompt (Lewis et al., 2020). This method can be broken down into three ways for storing dialogue history: each speaker's utterance individually, the entire session, and a summarized version of each session (denoted as *Utterance*, *Session Entire*, and *Session Sum.* in Table 3). The retrieval from the memory was performed using BM25 (Robertson et al., 2009) and cosine similarity with the OpenAI embeddings (OpenAI, 2024b).

For the LLMs, we used both API-based models (*i.e.*, Gemini-2.5 Flash (Google, 2025a), Gemini-2.0 Flash (Google, 2025b), GPT-4o-mini (OpenAI, 2024a), and GPT-4.1-nano (OpenAI, 2025)) and open-source models (*i.e.*, Llama 3.1-8B (Meta, 2024a), Llama 3.3-70B (Meta, 2024b), Mistral-

Table 3: The performance of the agents on Friends dialogue in DialSim. We conducted experiments three times and reported the accuracy and standard deviations. **Bold** indicates the best performance per retrieval method. Underline indicates the highest score for each model.

| Type | Model | Base LLM | RAG-based | | | | | |
| | | | BM25 | | | OpenAI Embedding | | |
| | | | Utterance | Session Entire | Session Sum. | Utterance | Session Entire | Session Sum. |
|---|---|---|---|---|---|---|---|---|
| API | GPT-4o-mini | 48.11 (1.26) | **35.46 (0.91)** | 52.11 (0.52) | 44.15 (2.35) | 41.29 (0.30) | **45.47 (2.08)** | 42.78 (1.03) |
| | GPT-4.1-nano | 26.01 (0.42) | 25.54 (1.13) | 31.03 (2.36) | 25.42 (1.48) | 29.50 (2.50) | 27.63 (0.97) | 26.44 (0.58) |
| | Gemini 2.5 Flash | 53.94 (1.26) | 30.86 (0.97) | **52.92 (1.66)** | 24.27 (2.01) | 37.16 (0.45) | 42.40 (1.15) | 23.75 (1.57) |
| | Gemini 2.0 Flash | 48.15 (0.68) | 29.46 (2.03) | 51.02 (1.85) | 23.67 (0.16) | 35.93 (0.42) | 42.02 (2.00) | 22.35 (0.91) |
| Open | Llama 3.3-70B | 34.65 (1.58) | 27.80 (4.43) | 44.91 (2.31) | 37.63 (0.69) | 38.70 (1.75) | 39.63 (1.51) | **45.30 (1.14)** |
| | Llama 3.1-8B | 34.31 (3.57) | 24.56 (0.87) | 38.23 (1.70) | 30.44 (1.31) | 24.61 (1.12) | 37.21 (0.95) | 32.95 (0.58) |
| | Mixtral-8x7B | 38.65 (0.37) | 26.82 (0.51) | 37.87 (2.36) | 31.10 (0.57) | 30.35 (2.14) | 32.61 (2.42) | 26.99 (0.47) |
| | Mistral-7B | 28.48 (0.58) | 27.71 (0.89) | 37.85 (1.01) | 34.31 (0.26) | 30.01 (1.20) | 34.99 (0.63) | 32.61 (1.42) |
| | Qwen 3-32B | 40.95 (3.35) | 30.82 (0.49) | 47.55 (1.29) | 36.40 (1.36) | 36.44 (1.92) | 42.87 (0.78) | 44.66 (0.42) |
| | Qwen 3-8B | 28.18 (1.25) | 30.01 (0.58) | 40.19 (1.66) | 35.55 (2.48) | 35.25 (1.01) | 36.78 (1.09) | 34.70 (0.57) |
| | Qwen 2.5-14B | 12.39 (0.28) | 21.75 (1.78) | 34.40 (2.48) | 26.69 (0.45) | 24.35 (1.93) | 30.57 (3.21) | 25.97 (2.29) |
| | Qwen 2.5-7B | 12.43 (1.33) | 24.14 (0.81) | 27.93 (1.16) | 22.22 (0.99) | 26.82 (0.73) | 29.29 (1.85) | 23.84 (1.00) |
| | Phi 4-14B | 22.86 (1.54) | 24.44 (1.44) | 33.25 (1.67) | 30.10 (0.94) | 30.95 (1.66) | 28.69 (1.19) | 29.50 (0.38) |
| | Phi 4 mini-3.8B | 10.47 (0.85) | 25.29 (2.08) | 31.16 (1.78) | 20.99 (0.99) | 30.95 (0.63) | 24.27 (2.26) | 19.75 (0.39) |
| | OLMo 2-7B | 24.61 (1.33) | 23.37 (0.68) | 13.11 (0.87) | 24.52 (4.27) | 30.35 (0.37) | 27.16 (1.12) | 28.86 (1.01) |
| | OLMoE-1B-7B | 17.97 (0.68) | 25.54 (2.18) | 6.90 (0.68) | 17.41 (6.67) | 31.42 (0.28) | 23.58 (2.35) | 27.71 (0.38) |
| | Tülu 3-8B | 24.48 (1.09) | 25.59 (0.97) | 32.10 (1.17) | 30.23 (0.47) | 32.35 (0.52) | 31.59 (0.69) | 29.63 (0.10) |
| | Random Guessing | 20.00 | 20.00 | 20.00 | 20.00 | 20.00 | 20.00 | 20.00 |

7B (Jiang et al., 2023), Mixtral-8x7B (Jiang et al., 2024), Qwen 3-32B, Qwen 3-8B (Yang et al., 2025), Qwen 2.5-14B, Qwen 2.5-7B (Yang et al., 2024), Phi 4-14B, Phi 4 mini-3.8B (Abdin et al., 2024)), OLMo 2-7B (OLMo et al., 2024), OLMoE-1B-7B (Muennighoff et al., 2024), and Tülu 3-8B (Lambert et al., 2024)). To emulate conversational settings for the open-source models, we constructed chat-style prompts by applying templates using the Hugging Face `apply_chat_template` method.[6]

## 5.3 RESULTS

**Overall Performance** Table 3 shows that API-based models outperformed others, likely due to their superior inference capabilities. However, all baseline performances remained below 60%, indicating that current LLMs face substantial limitations when serving as conversational agents in long-term multi-party dialogue settings. Similar trends were observed across the Friends, The Big Bang Theory, and The Office datasets, with detailed results provided in Appendix H.

**Extended context windows alone are insufficient for long-term dialogue understanding.** As shown in Table 3, models such as GPT-4o-mini, GPT-4.1-nano, and Gemini 2.0 Flash, despite supporting context lengths ranging from 128k to 1M tokens, performed worse than the best retrieval-augmented method, BM25-Session Entire. Only Gemini 2.5 Flash, equipped with both a 1M-token context window and strong reasoning capabilities, achieved the highest overall accuracy of 53.94% under Base LLM setting. These findings suggest that simply increasing the context window is not enough; models must also possess robust reasoning and comprehension capabilities to manage long-term conversations effectively.

**Among RAG-based methods, storing the entire session consistently outperforms other history storing methods.** As illustrated in Table 3, storing the entire session yields better performance than storing individual utterances or using summarization. This is likely because individual utterances lack sufficient context, and summarization may omit critical information. Moreover, models with strong long-term dialogue understanding (*e.g.*, Gemini 2.5 Flash, Mixtral-8x7B) tended to achieve higher Base-LLM scores, whereas models with strong summarization capabilities (*e.g.*, Llama 3.3-70B) performed best when using the Session Summary method.

**Effective memory management is critical for RAG-based agents engaging in long-term multi-party dialogues.** We further evaluated model performance in an oracle setting, where agents were

---

[6]https://huggingface.co/docs/transformers/en/chat_templating

Table 4: Model performance under different character name settings: (1) using anonymized generic names, (2) using the original character names without anonymization, and (3) swapping the character names (adversarial setting). Performance improves when original names are used and decreases when names are swapped. The full experimental results are provided in Appendix J.

| Type | Model | BM25-Session Entire | | | OpenAI Embedding-Session Entire | | |
|------|-------|------------------|----------|----------------|-----------------|----------|----------------|
| | | Generic Names | Original | Swapping Names | Generic Names | Original | Swapping Names |
| API | GPT-4o-mini | 52.11 | 55.62 (↑ 3.51) | 45.08 (↓ 7.03) | 45.47 | 52.81 (↑ 7.34) | 41.38 (↓ 4.09) |
| | Gemini 2.5 Flash | 52.92 | 56.26 (↑ 3.34) | 51.40 (↓ 1.52) | 42.40 | 50.19 (↑ 7.79) | 43.23 (↑ 0.83) |
| | Gemini 2.0 Flash | 51.02 | 54.34 (↑ 3.32) | 47.81 (↓ 3.21) | 42.02 | 45.72 (↑ 3.70) | 38.53 (↓ 3.49) |
| Open | Llama3.3-70B | 44.91 | 48.23 (↑ 3.32) | 42.87 (↓ 2.04) | 39.63 | 43.51 (↑ 3.88) | 38.61 (↓ 1.02) |
| | Mixtral-8x7B | 37.87 | 46.47 (↑ 8.60) | 37.72 (↓ 0.15) | 32.61 | 41.24 (↑ 8.63) | 31.80 (↓ 0.81) |
| | Qwen 3-32B | 47.55 | 51.72 (↑ 4.17) | 46.32 (↓ 1.23) | 42.87 | 45.72 (↑ 2.85) | 39.04 (↓ 3.83) |
| | Average | 47.73 | 52.11 (↑ 4.38) | 45.20 (↓ 2.53) | 40.83 | 46.53 (↑ 5.70) | 38.77 (↓ 2.06) |

provided with evidence sessions along with timestamps (see Figure 2). As shown in Figure 4, performance in the oracle setting was 10–30% higher compared to that of the best memory management method. This substantial performance gain emphasizes the importance of advanced history storage and retrieval techniques. The complete results of the oracle experiments are provided in Appendix I.

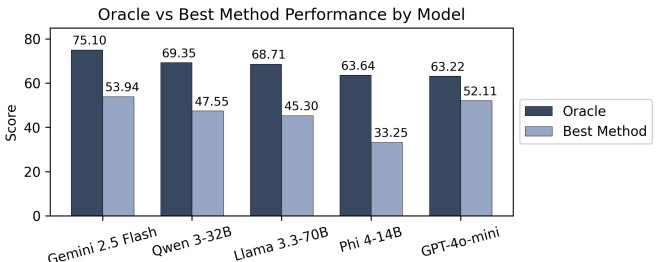

Figure 4: The performance comparison between the oracle setting and the best memory management method.

**TKG-based questions are more challenging than fan quiz-style questions, with two-hop reasoning posing particular difficulty.** To assess the difficulty levels across different question types, we conducted an error analysis on Gemini-2.5 Flash Base LLM, which showed the highest performance. The results showed that fan quiz-based questions had an accuracy of 62.15%, while TKG-based questions scored lower at 50.83%, highlighting the greater difficulty of TKG-based questions. Breaking down TKG-based questions further, one-hop questions had a performance of 69.19%, whereas two-hop questions had a performance of 19.28%, underscoring the challenge of two-hop questions. Furthermore, even in the oracle setting, while the performance of one-hop questions increased to 83.60%, two-hop questions remained at 51.74%. This indicates that two-hop questions are challenging not only in terms of history retrieval but also in reasoning across the given sessions.

**Character anonymization in LongDialQA is essential for fair evaluation of conversational agents.** We conducted additional experiments using DialSim on both the original version of LongDialQA (without character anonymization) and the adversarial version where character names were swapped (*e.g.*, Joey ↔ Monica). As shown in Table 4, performance improved in the original setting, likely because models leveraged pre-trained knowledge alongside dialogue history. These results support the necessity of name anonymization to ensure reliable evaluation. Additionally, performance declined when character names were swapped. This suggests that dialogues with generic names introduce new information, whereas swapped names conflict with pre-trained knowledge, leading to reduced reasoning performance. Detailed results are provided in Appendix J.

## 6 CONCLUSION

In this paper, we introduce DialSim, a simulator designed for evaluating the ability of conversational agents to understand long-term, multi-party dialogues. To run DialSim, we first constructed LongDialQA, a dataset based on dialogues from well-known TV show scripts. LongDialQA also includes questions derived from fan quizzes and a temporal knowledge graph, enabling a comprehensive assessment of the agents. Using DialSim, we evaluated the latest conversational agents and uncovered significant limitations in handling complex, multi-party, long-term dialogues. We hope our work paves the way for more rigorous and realistic evaluation standards in conversational AI.

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

## A  LLM'S PRIOR KNOWLEDGE OF THE TV SHOWS

We asked GPT-4o to explain the plot of specific episodes of Friends. It accurately described the plots, as shown in Figure 5, 6. Notably, it provided these answers without any web browsing, suggesting that GPT-4o might have learned about these TV shows during its pre-training process.

## B  DATE ASSIGNMENT

We first extracted elements from the scripts that could indicate dates (*e.g.*, Valentine's Day, Christmas Eve). Then, we reviewed the scripts again to analyze the relative timing of the sessions. For example, if there is a line mentioning that Chandler broke up with his girlfriend two days ago, we

annotated the session where he broke up with his girlfriend as occurring two days prior to the mentioned session. Next, while watching each episode, we pinpointed sessions where the dates might have changed by observing whether the characters' outfits changed between sessions. Finally, we assigned a specific date to each session based on the actual broadcast date of the episode, adjusting for the relative differences in dates and events such as Christmas.

## C    QUESTION GENERATION BASED ON FAN QUIZZES

For each scene $s_{i,k}$ from episode $p_i$ in $Script_{pre}$, we define the set of answerable questions as $FanA_{i,k}$ and the set of unanswerable questions as $FanU_{i,k}$. The process of generating questions based on fan quizzes is as follows.

First, we collected quizzes for each season and episode of Friends, The Big Bang Theory, and The Office from the FunTrivia website. For each episode $p_i$ in $Script_{pre}$, we used GPT-4 to determine if the crawled questions $CrQ_i = \{q_{i,0}, q_{i,1}, ..., q_{i,l}\}$ could be answered using only $p_i$. If a question $q_{i,m}$ could be answered, GPT-4 identified the scenes $ES_{i,m}$ that provide evidence for the answer, compiling them into $Q_i = \{(q_{i,m}, ES_{i,m})\}_{m=0}^{l}$. Subsequently, the authors reviewed each $ES_{i,m}$, made necessary corrections, and annotated whether a single scene from $ES_{i,m}$ was sufficient to answer $q_{i,m}$ or if multiple scenes were needed to be considered simultaneously. For each $s_{i,k}$ within $p_i$, we assessed the answerability of the questions in $Q_i$.

For each $s_{i,k}$, if a question $q_{i,m}$ could be answered using just one scene, and $s_{i,k}$ occurs after the initial appearance of the main character in $ES_{i,m}$, we included $q_{i,m}$ in $FanA_{i,k}$. This ensures that the main character had adequate exposure to the relevant evidence. Additionally, for questions requiring verification across multiple scenes, if the main character appears in all $ES_{i,m}$ scenes and $s_{i,k}$ occurs after the last scene of $ES_{i,m}$, we included $q_{i,m}$ in $FanA_{i,k}$. If the main character does not appear in any of the $ES_{i,m}$ scenes, $q_{i,m}$ was included in $FanU_{i,k}$ since the main character has not experienced any evidence to answer the question. The rest are not included in the dataset as it is unclear whether they are answerable per scene. Additionally, to generate questions that require long-term memory, we added the most recent date of the evidence scenes for each question.

## D    QUESTION GENERATION BASED ON A TEMPORAL KNOWLEDGE GRAPH

### D.1    RELATIONS

We used the following 32 relations: 'age', 'alumni', 'boss', 'boyfriend', 'brother', 'client', 'date of birth', 'dating with', 'ex-boyfriend', 'ex-fiance', 'ex-fiancee', 'ex-girlfriend', 'ex-husband', 'ex-roommate', 'ex-wife', 'father', 'fiance', 'fiancee', 'girlfriend', 'hometown', 'husband', 'job', 'major', 'mother', 'neighbor', 'pet', 'place of birth', 'place of work', 'roommate', 'sister', 'subordinate', 'wife'.

### D.2    QUESTION TEMPLATES AND GENERATED QUESTIONS

Templates for one-hop questions are provided in Table 5 and Table 6. The former contains templates without temporal information, while the latter includes templates with temporal details. Since relations like "brother" and "sister" remain constant over time, questions about these relations do not require temporal information. Hence, no temporal templates were created for them. In Table 6, "on {time}" is used, but {time} can be not only the full date (year, month, and day) but also just the year and month, or even just the year. In these cases, "in {time}" is used.

The templates for two-hop questions are available in Table 7. These templates incorporate temporal information. To frame questions in the present tense, adjust the verbs to the present tense and remove the temporal information, following the approaches demonstrated in Table 5 and Table 6.

## E    CHARACTER STYLE TRANSFER

Table 8 shows the results of the character style transfer for three selected questions. To make the questions sound more natural and conversational, we prepended each one with "By the way,". This

helps them blend seamlessly into the flow of the conversation. The table shows how each question appears when rephrased in the style of various characters. The 'Default' setting is applied when the question is asked by a character who is not a recurring character of the TV show.

## F    PROMPT FOR RESPONSE GENERATION

The prompt given to the conversational agent to answer questions using dialogue history is shown in Table 9. An example where the placeholders from Table 9 are filled with actual values can be found in Table 10.

## G    EXPERIMENTAL SETTING

### G.1    QUESTION FORMAT

LongDialQAis a dataset that includes pairs of questions, answers, and choices. The questions are available in three formats: template-based multiple-choice, natural language multiple-choice, and open-ended. Users can choose any of these formats to evaluate the agent's performance.

First, we provide multiple-choice questions in both template and natural language formats. For example, a template-based question might be, "Who was going out with Paul in September 1994?" with choices "(A) Emily, (B) Monica, (C) Ryan, (D) Rachel, (E) I don't know". In contrast, the same question in natural language format could be phrased as, "Who was going out with Paul in September 1994? Was it Emily, Monica, Ryan, Rachel, or do you not know?"

Additionally, we offer the option to ask questions in an open-ended format (*e.g.*, "Who was going out with Paul in September 1994?") without providing answer choices. This approach allows us to evaluate the agent's ability to generate open-ended responses. The open-ended format is particularly useful for fan quiz-based questions, where some answers may require longer responses (*e.g.*, Question: "Why did Monica and Chandler say they were late getting to the hospital?" Correct answer: "Monica went back for her jacket").

For natural language multiple-choice and open-ended questions, a response is considered correct if it exactly matches the correct answer. If the response does not match exactly, the score is determined by comparing the response with the correct answer using a different language model (*i.e.*, GPT-4o mini).

### G.1.1    CHOICES IN MULTIPLE-CHOICE QUESTIONS

The number of questions based on fan quizzes was significantly smaller than the questions based on the TKG. Thus, 30% of the questions were intentionally extracted from the fan quiz-based during the simulation. Since each question has five choices, unanswerable questions were set to comprise 20% of the total to fairly stratify the correct answers.

### G.2    NUMBER OF RETRIEVED DIALOGUE HISTORY

By default, agents retrieved up to 20 utterances, 10 entire sessions, and 15 session summaries, depending on the storing method, though some LLMs with shorter context lengths retrieved fewer histories accordingly.

## H    EXPERIMENTAL RESULTS FOR THE BIG BANG THEORY AND THE OFFICE

The experimental results for The Big Bang Theory and The Office are provided in Table 11 and Table 12, respectively.

## I  EXPERIMENTAL RESULTS IN THE ORACLE SETTING

Figure 7 shows the performance comparison between the BM25-Session Entire setting and the Oracle setting. Gemini-2.5 Flash achieved the highest performance with a score of 75.10% in the Oracle setting.

## J  EXPERIMENTAL RESULTS ON ADVERSARIAL TEST

In the adversarial test, we altered the characters' names and ran experiments under different conditions. Table 13 presents the results of original character names. Table 14 displays the results when characters' names were mixed.

## K  ANNOTATOR INSTRUCTIONS

Figure 8 and Figure 9 show the screenshots of the dataset labeling process. Figure 8 illustrates the annotation process for the questions based on fan quizzes, and Figure 9 describes the review process for selecting triples for the TKG.

Figure 5: The result of asking GPT-4o to explain Season 2, Episode 7 of Friends.

Tell me the plot of episode 14 from season 3 of the TV show Friends.

Episode 14 of season 3 of the TV show "Friends" is titled "The One with Phoebe's Ex-Partner." Here is a summary of the plot:

Phoebe's former singing partner, Leslie, shows up and wants to get back together. Initially, Phoebe is excited about the reunion, but their rekindled partnership quickly hits a snag when Leslie sells out by using "Smelly Cat" in a commercial. This betrayal causes Phoebe to sever ties with Leslie once again.

Meanwhile, Chandler dates a woman named Ginger who has a wooden leg, a fact he only discovers after Ross inadvertently tells him. This revelation makes Chandler uncomfortable, leading to some awkward and humorous situations. Joey also remembers Ginger from a previous encounter where he accidentally threw her artificial leg into a fire.

Ross is dealing with relationship issues with Rachel. They have a fight over a list Ross made, comparing her to Julie, his previous girlfriend. Despite this, Ross tries to make amends by taking her to a planetarium. However, their attempt at romance is interrupted by a loud child in the audience, and Rachel's feelings remain unresolved.

The episode balances the comedic elements with more serious moments, showcasing the dynamics between the friends and their relationships.

Figure 6: The result of asking GPT-4o to explain Season 3, Episode 14 of Friends.

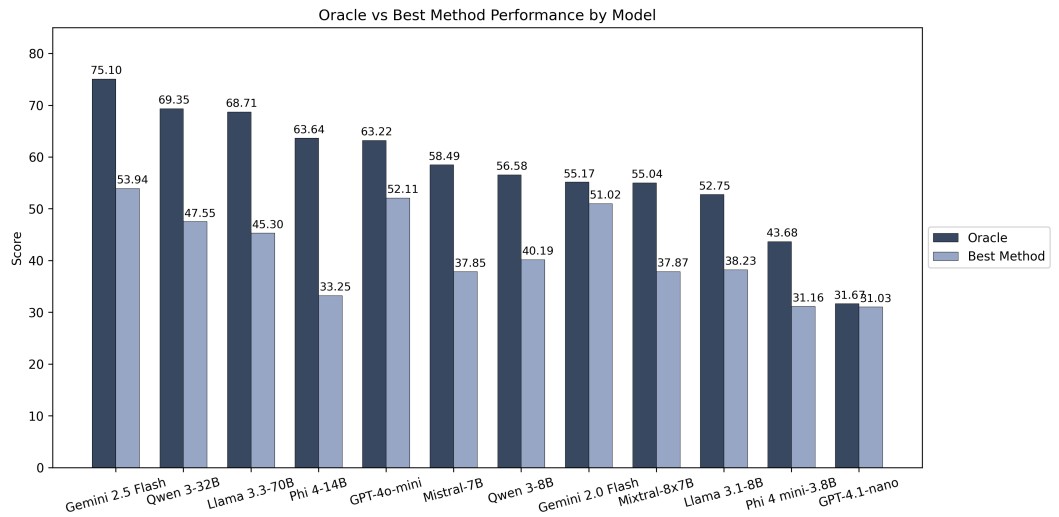

Figure 7: The performance comparison between the Oracle setting and the best memory management method.

Table 5: Templates for one-hop questions without temporal information.

| Question Type | Relation | Template | Question Example |
|---|---|---|---|
| | alumni | Who is {sub}'s alumni? | Who is Lincoln High School's alumni? |
| | boss | Who is {sub}'s boss? | Who is Chandler's boss? |
| | subordinate | Who is {sub}'s subordinate? | Who is Chandler's subordinate? |
| | client | Who is {sub}'s client? | Who is Chandler's client? |
| | neighbor | Who is {sub}'s neighbor? | Who is Chandler's neighbor? |
| | roommate | Who is {sub}'s roommate? | Who is Chandler's roommate? |
| | ex-roommate | Who is {sub}'s ex-roommate? | Who is Chandler's ex-roommate? |
| | fiance | Who is {sub}'s fiance? | Who is Rachel's fiance? |
| | fiancee | Who is {sub}'s fiancee? | Who is Ross's fiancee? |
| | ex-fiance | Who is {sub}'s ex-fiance? | Who is Rachel's ex-fiance? |
| | ex-fiancee | Who is {sub}'s ex-fiancee? | Who is Ross's ex-fiancee? |
| | pet | Who is {sub}'s pet? | Who is Ross's pet? |
| | dating with | Who is dating {sub}? | Who is dating Ross? |
| | job | What is {sub}'s job? | What is Ross's job? |
| | place of work | Where does {sub} work? | Where does Ross work? |
| Without Time | age | How old is {sub}? | How old is Ross? |
| | major | What is {sub}'s major? | What is Ross's major? |
| | mother | Who is {sub}'s mother? | Who is Ross's mother? |
| | father | Who is {sub}'s father? | Who is Ross's father? |
| | place of birth | Where was {sub} born? | Where was Ben born? |
| | hometown | Where is {sub}'s hometown? | Where is Monica's hometown? |
| | date of birth | When was {sub} born? | When was Ben born? |
| | husband | Who is {sub}'s husband? | Who is Emily's husband? |
| | wife | Who is {sub}'s wife? | Who is Ross's wife? |
| | girlfriend | Who is {sub}'s girlfriend? | Who is Joey's girlfriend? |
| | boyfriend | Who is {sub}'s boyfriend? | Who is Monica's boyfriend? |
| | ex-husband | Who is {sub}'s ex-husband? | Who is Carol's ex-husband? |
| | ex-wife | Who is {sub}'s ex-wife? | Who is Ross's ex-wife? |
| | ex-girlfriend | Who is {sub}'s ex-girlfriend? | Who is Ross's ex-girlfriend? |
| | ex-boyfriend | Who is {sub}'s ex-boyfriend? | Who is Rachel's ex-boyfriend? |
| | brother | Who is {sub}'s brother? | Who is Monica's brother? |
| | sister | Who is {sub}'s sister? | Who is Ross's sister? |

Table 6: Templates for one-hop questions with temporal information.

| Question Type | Relation | Template | Question Example |
|---|---|---|---|
| | boss | Who was {sub}'s boss on {time}? | Who was Chandler's boss on September 26th, 1994? |
| | client | Who was {sub}'s client on {time}? | Who was Chandler's client on September 26th, 1994? |
| | neighbor | Who was {sub}'s neighbor on {time}? | Who was Chandler's neighbor on September 26th, 1994? |
| | roommate | Who was {sub}'s roommate on {time}? | Who was Chandler's roommate on September 26th, 1994? |
| | fiance | Who was {sub}'s fiance on {time}? | Who was Rachel's fiance on September 26th, 1994? |
| | fiancee | Who was {sub}'s fiancee on {time}? | Who was Ross's fiancee on September 26th, 1994? |
| | pet | Who was {sub}'s pet on {time}? | Who was Ross's pet on September 26th, 1994? |
| With Time | dating with | Who dated {sub} on {time}? | Who dated Ross on September 26th, 1994? |
| | job | What was {sub}'s job on {time}? | What was Monica's job on September 26th, 1994? |
| | place of work | Where did {sub} work on {time}? | Where did Monica work on September 26th, 1994? |
| | age | How old was {sub} on {time}? | How old was Monica on September 26th, 1994? |
| | major | What was {sub}'s major on {time}? | What was Ross's major on September 26th, 1994? |
| | husband | Who was {sub}'s husband on {time}? | Who was Emily's husband on September 26th, 1994? |
| | wife | Who was {sub}'s wife on {time}? | Who was Ross's wife on September 26th, 1994? |
| | girlfriend | Who was {sub}'s girlfriend on {time}? | Who was Ross's girlfriend on September 26th, 1994? |
| | boyfriend | Who was {sub}'s boyfriend on {time}? | Who was Rachel's boyfriend on September 26th, 1994? |

Table 7: Templates for two-hop questions.

| First Relation | Second Relation | Template | Question Example |
|---|---|---|---|
| roommate, wife, husband, girlfriend, boyfriend, client, neighbor, boss, subordinate, fiance, fiancee | roommate, wife, husband, pet, girlfriend, boyfriend, client, neighbor, boss, subordinate, fiance, fiancee | {sub1} had a {First Relation} on {time1}. Who was the {Second Relation} of the {First Relation} on {time2}? | Monica had a roommate on September 26th, 1994. Who was the boyfriend of the roommate on October 5th, 1996? |
| | dating with | {sub1} had a {First Relation} on {time1}. Who dated the {First Relation} on {time2}? | Monica had a roommate on September 26th, 1994. Who dated the roommate on October 5th, 1996? |
| | job, major, age | {sub1} had a {First Relation} on {time1}. What was the {Second Relation} of the {First Relation} on {time2}? | Monica had a roommate on September 26th, 1994. What was the job of the roommate on October 5th, 1996? |
| | mother, father, son, daughter, sister, brother | {sub1} had a {First Relation} on {time1}. Who is the {Second Relation} of the {First Relation}? | Monica had a roommate on September 26th, 1994. Who is the mother of the roommate? |
| | date of birth, place of birth, | {sub1} had a {First Relation} on {time1}. When (Where) was the {First Relation} born? | Monica had a roommate on September 26th 1994. When was the roommate born? |
| | place of work | {sub1} had a {First Relation} on {time1}. Where did the {First Relation} work on {time2}? | Monica had a roommate on September 26th, 1994. Where did the roommate work on October 5th, 1996? |
| | hometown | {sub1} had a {First Relation} on {time1}. Where is the hometown of the {First Relation}? | Monica had a roommate on September 26th, 1994. Where is the hometown of the roommate? |
| dating with | roommate, wife, husband, girlfriend, boyfriend, client, neighbor, boss, subordinate, fiance, fiancee | {sub1} dated a person on {time1}. Who was the {Second Relation} of the person on {time2}? | Monica dated a person on September 26th, 1994. Who was the boss of the person on October 5th, 1996? |
| mother, father, son, daughter, sister, brother | roommate, wife, husband, girlfriend, boyfriend, client, neighbor, boss, subordinate, fiance, fiancee | Who was the {Second Relation} of {sub1}'s {First Relation} on {time2}? | Who was the roommate of Ross's sister on September 26th, 1994? |
| | dating with | Who dated {sub1}'s {First Relation} on {time2}? | Who dated Ben's father on September 26th, 1994? |
| | job, age, major | What was the {Second Relation} of {sub1}'s {First Relation} on {time2}? | What was the job of Ben's father on September 26th, 1994? |
| | mother, father, son, daughter, sister, brother | Who is the {Second Relation} of {sub1}'s {First Relation}? | Who is the mother of Ross's son? |
| | date of birth, place of birth | When (Where) was {sub1}'s {First Relation} born? | When was Monica's brother born? |
| | place of work | Where did {sub1}'s {First Relation} work on {time2}? | Where did Monica's brother work on October 5th, 1996? |
| | hometown | Where is the hometown of {sub1}'s {First Relation}? | Where is the hometown of Ross's son? |

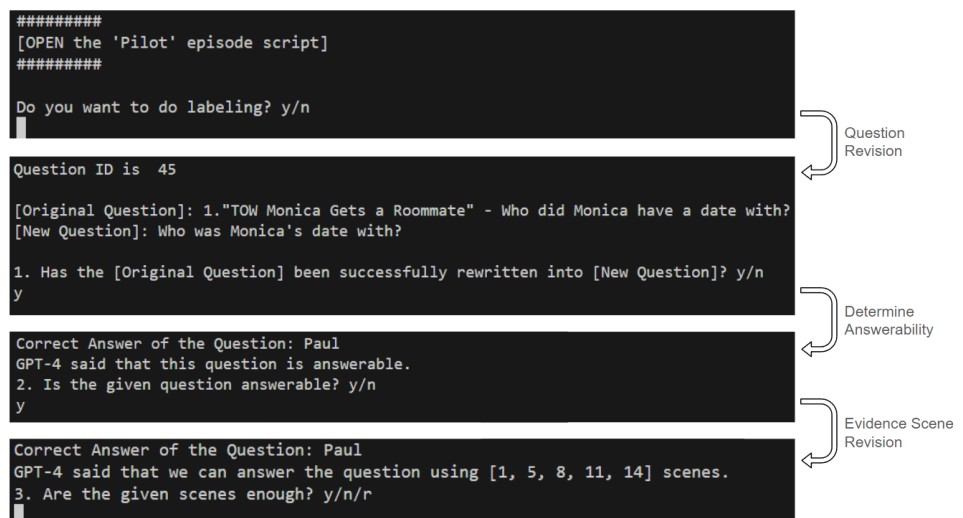

Figure 8: The actual process of annotating questions from fan quizzes.

Table 8: Examples of the results of character style transfer.

| Original Question | Character | Style Transferred Question |
|---|---|---|
| By the way, how did Rachel buy her new boots? | Default | Hey, any idea what Rachel used to snag those stylish new boots of hers? |
| | Monica | Hey, do you know what Rachel used to snag those super cute new boots she's been rocking? |
| | Chandler | So, could we BE any more curious about how Rachel snagged those new boots? |
| | Joey | Hey, how did Rachel manage to snag those killer boots, huh? |
| | Phoebe | Oh my gosh! Do you have any idea how Rachel snagged those super cute new boots? |
| By the way, who dated Monica on September 22, 1994? | Default | So, who was Monica's date on the night of September 22, 1994? |
| | Chandler | Oh, could you BE any more specific about who was going out with Monica on September 22, 1994? |
| | Joey | Hey, just outta curiosity, who was goin' out with Monica on September 22, 1994? |
| | Phoebe | Oh my gosh, so like, who was Monica's date on that super specific day, September 22, 1994? |
| | Rachel | Oh my god, so like, who was going out with Monica on September 22, 1994?' |
| By the way, Rachel had a roommate on October 28, 1994. Who dated the roommate in September 1994? | Default | Oh. My. God. Remember when Rachel had a roommate back on October 28, 1994? So, who was going out with that roommate by September 1994? |
| | Monica | Hey, just out of curiosity, do you know who was going out with Rachel's roommate from back in September 1994? I remember she got that roommate around October 28, 1994. |
| | Chandler | So, just for a little stroll down memory lane, Rachel was bunking with someone on October 28, 1994. Any wild guesses on who was dating this mystery co-habitant by September 1994? |
| | Joey | Hey, so you know how Rachel was living with someone back on October 28, 1994, right? So I'm just wonderin' here, who was going out with this roommate of hers in September 1994? |
| | Phoebe | By the way, Rachel had a roommate on October 28, 1994. Who dated the roommate in September 1994? |

```
#########
[OPEN the 'Monica Gets A Roommate' episode script]
#########

Scene number list is  [1]

0 ['phoebe', 'ex-boyfriend', 'carl']
1 ['ross', 'ex-spouse', 'carol']
2 ['monica', 'brother', 'ross']
3 ['rachel', 'fiance', 'barry']
4 ['rachel', 'ex-fiance', 'barry']
5 ['monica', 'friend', 'rachel']
6 ['rachel', 'visited place', "monica's building"]
7 ['rachel', 'hometown', 'the city']

Which triples can be survived? (e.g. int, int, int)
```

Figure 9: The actual process of reviewing extracted triples.

Table 9: In the <<<Chatbot>>> placeholder, the name of the main character (*i.e.*, Ross, Sheldon, Michael) for each TV show is inserted. In the <<<Date>>> placeholder, the date of the session in which the question is being asked is inserted. In the <<<Dialog_History>>> placeholder, the dialogue history that the agent will use is inserted. In the <<<Question>>> placeholder, the question that the agent should answer along with five choices is inserted.

| Prompt for Response Generation |
|---|
| You are <<<Chatbot>>>, a long-term conversational agent capable of interacting with multiple users. |
| Based on the [Retrieved Dialogue History] provided, please answer the given [Question]. |
| Note the following points: |
| 1. Your answer must exclusively be one of the options: (A), (B), (C), (D), (E). |
| 2. Your responses should solely rely on the retrieved dialogue history. |
| 3. This question is being asked in the context of <<<Date>>>. |
| |
| [Retrieved Dialogue History] |
| <<<Dialog_History>>> |
| [Question] <<<Question>>> |
| [Answer] |

Table 10: An actual example of the prompt for response generation.

| Prompt for Response Generation |
|---|
| You are Ross, a long-term conversational agent capable of interacting with multiple users. |
| Based on the [Retrieved Dialogue History] provided, please answer the given [Question]. |
| Note the following points: |
| 1. Your answer must exclusively be one of the options: (A), (B), (C), (D), (E). |
| 2. Your responses should solely rely on the retrieved dialogue history. |
| 3. This question is being asked in the context of [February 26, 1999]. |
| |
| [Retrieved Dialogue History] |
| |
| [Session #1 on September 22, 1994] |
| <<Session Omitted>> |
| Ross: No, go on! It's Paul the Wine Guy! |
| Phoebe: What does that mean? Does he sell it, drink it, or just complain a lot? |
| Monica: Hi, come in! Paul, this is.. ... everybody, everybody, this is Paul. |
| All: Hey! Paul! Hi! The Wine Guy! Hey! |
| Chandler: I'm sorry, I didn't catch your name. Paul, was it? |
| Monica: Okay, umm-umm, I'll just–I'll be right back, I just gotta go ah, go ah. . . |
| Ross: A wandering? |
| Monica: Change! Okay, sit down. Two seconds. |
| Phoebe: Ooh, I just pulled out four eyelashes. That can't be good. |
| <<Session Omitted>> |
| |
| [Session #2 on May 20, 1998] |
| <<Session Omitted>> |
| Rachel: Umm, hi! |
| Ross: Hi. |
| Rachel: Is Monica around? I-I have to ask her something. |
| Ross: She's doing her laundry. |
| <<Session Omitted>> |
| Rachel: Y'know what Ross? You're not going anywhere. You're gonna sit right here. |
| I'm gonna make you a cup of tea and we're gonna talk this thing whole out. All right? Hey, Dave! |
| Dave: Yeah? |
| Rachel: Umm, listen, I'm gonna need to take a rain check, my roommate is just really sick. |
| Okay? Bye! Honey, listen, I know, I know things seem so bad right now. |
| |
| [Question] Chandler: So, just for a little stroll down memory lane, Rachel was bunking with someone in May 1998. |
| Any wild guesses on who was dating this mystery cohabitant by September 22, 1994? |
| (A) Paolo (B) Paul (C) Roger (D) Vince (E) I don't know. |
| [Answer] |

Table 11: The performance of the agents on The Big Bang Theory dialogue in DialSim. We conducted experiments three times and reported the accuracy and standard deviations. **Bold** indicates the best performance per retrieval method.

| Type | Model | Base LLM | RAG-based | | | | | |
| | | | BM25 | | | OpenAI Embedding | | |
| | | | Utterance | Session Entire | Session Sum. | Utterance | Session Entire | Session Sum. |
|---|---|---|---|---|---|---|---|---|
| API | GPT-4o-mini | 41.33 (0.93) | 21.95 (0.95) | **52.54 (1.65)** | 36.88 (1.48) | **29.72 (2.48)** | **41.69 (0.68)** | **42.29 (2.22)** |
| | GPT-4.1-nano | 13.45 (0.38) | 8.74 (1.24) | 18.44 (0.53) | 9.46 (0.66) | 11.93 (0.39) | 15.11 (1.69) | 13.38 (1.12) |
| | Gemini 2.5 Flash | **56.15 (5.77)** | 13.39 (1.73) | 46.40 (1.03) | 26.63 (2.45) | 26.33 (0.91) | 34.66 (1.43) | 29.59 (2.76) |
| | Gemini 2.0 Flash | 44.62 (0.86) | 13.89 (1.13) | 45.07 (1.27) | 21.43 (2.96) | 24.34 (1.82) | 36.39 (1.25) | 24.32 (5.92) |
| Open | Llama 3.3-70B | 30.91 (0.02) | 15.36 (0.84) | 41.72 (1.07) | 31.03 (4.27) | 29.33 (0.43) | 34.53 (2.71) | 36.92 (0.91) |
| | Llama 3.1-8B | 32.99 (1.87) | 13.35 (1.29) | 31.06 (2.27) | 24.31 (0.97) | 25.32 (0.73) | 30.21 (1.92) | 35.98 (1.69) |
| | Mixtral-8x7B | 38.29 (1.37) | **24.06 (1.63)** | 50.16 (1.46) | **39.83 (1.55)** | 29.61 (0.76) | 41.41 (1.42) | 33.08 (2.15) |
| | Mistral-7B | 22.56 (1.94) | 19.39 (0.16) | 38.16 (1.55) | 29.58 (1.67) | 28.15 (1.50) | 33.76 (0.93) | 37.55 (2.85) |
| | Qwen 3-32B | 34.86 (1.59) | 15.31 (1.07) | 48.45 (1.20) | 31.49 (1.25) | 29.37 (0.76) | 40.62 (1.34) | 38.29 (0.53) |
| | Qwen 3-8B | 17.25 (3.09) | 20.25 (0.59) | 43.26 (2.60) | 35.24 (1.39) | 25.41 (0.81) | 34.87 (2.32) | 31.50 (0.84) |
| | Phi 4-14B | 17.31 (0.91) | 10.12 (0.41) | 31.39 (1.02) | 21.90 (2.82) | 20.99 (1.13) | 22.10 (1.12) | 25.75 (2.01) |
| | Phi 4 mini-3.8B | 7.57 (0.32) | 15.48 (1.58) | 30.88 (0.82) | 21.62 (1.34) | 26.58 (2.34) | 21.50 (0.12) | 14.45 (0.59) |
| | Random Guessing | 20.00 | 20.00 | 20.00 | 20.00 | 20.00 | 20.00 | 20.00 |

Table 12: The performance of the agents on The Office dialogue in DialSim. We conducted experiments three times and reported the accuracy and standard deviations. **Bold** indicates the best performance per retrieval method.

| Type | Model | Base LLM | RAG-based | | | | | |
| | | | BM25 | | | OpenAI Embedding | | |
| | | | Utterance | Session Entire | Session Sum. | Utterance | Session Entire | Session Sum. |
|---|---|---|---|---|---|---|---|---|
| API | GPT-4o-mini | 47.49 (0.81) | **44.54 (0.52)** | **60.92 (0.40)** | **55.55 (2.20)** | 53.49 (1.93) | 59.57 (2.00) | **60.81 (1.03)** |
| | GPT-4.1-nano | 15.10 (0.76) | 15.04 (0.16) | 24.73 (0.15) | 15.62 (1.59) | 22.19 (0.60) | 23.60 (1.42) | 22.49 (0.88) |
| | Gemini 2.5 Flash | **64.87 (7.98)** | 23.83 (1.44) | 48.14 (0.62) | 30.28 (6.91) | 35.52 (0.59) | 48.86 (1.54) | 28.93 (6.25) |
| | Gemini 2.0 Flash | 58.91 (2.29) | 25.00 (0.44) | 50.40 (0.95) | 27.11 (1.98) | 38.03 (0.77) | 49.16 (0.45) | 33.38 (3.46) |
| Open | Llama 3.3-70B | 33.23 (0.76) | 21.43 (0.57) | 43.80 (0.41) | 35.47 (1.37) | 35.76 (0.67) | 47.12 (0.76) | 47.27 (1.25) |
| | Llama 3.1-8B | 12.15 (0.23) | 24.56 (1.33) | 44.29 (0.72) | 39.35 (3.09) | 35.71 (1.26) | 42.12 (0.14) | 41.79 (1.56) |
| | Mixtral-8x7B | 37.91 (1.32) | 31.30 (0.93) | 48.48 (0.43) | 41.56 (1.32) | 35.87 (0.35) | 45.81 (0.88) | 35.92 (1.03) |
| | Mistral-7B | 28.30 (0.03) | 26.46 (0.71) | 40.88 (1.21) | 37.58 (0.38) | 35.50 (0.69) | 39.78 (0.41) | 40.14 (0.45) |
| | Qwen 3-32B | 27.01 (1.61) | 22.18 (1.16) | 45.78 (1.10) | 38.22 (4.11) | 34.11 (1.45) | 47.25 (1.31) | 47.46 (0.81) |
| | Qwen 3-8B | 13.69 (0.40) | 21.89 (0.31) | 38.49 (0.89) | 33.12 (2.94) | 38.54 (1.21) | 36.37 (1.38) | 36.37 (0.90) |
| | Phi 4-14B | 16.38 (0.10) | 17.86 (1.17) | 28.28 (0.97) | 23.81 (2.26) | 25.65 (0.74) | 29.72 (0.89) | 35.14 (0.95) |
| | Phi 4 mini-3.8B | 10.92 (0.60) | 22.69 (1.36) | 24.43 (0.79) | 19.94 (1.05) | 36.12 (0.59) | 27.98 (1.24) | 27.84 (1.23) |
| | Random Guessing | 20.00 | 20.00 | 20.00 | 20.00 | 20.00 | 20.00 | 20.00 |

Table 13: The performance of the agents on original Friends dialogue in DialSim. We conducted experiments three times and reported the accuracy and standard deviations. **Bold** indicates the best performance per retrieval method.

| Type | Model | Base LLM | RAG-based | | | | | |
| | | | BM25 | | | OpenAI Embedding | | |
| | | | Utterance | Session Entire | Session Sum. | Utterance | Session Entire | Session Sum. |
|---|---|---|---|---|---|---|---|---|
| API | GPT-4o-mini | 50.28 (0.51) | **44.00 (2.62)** | 55.62 (0.45) | **48.66 (3.58)** | **47.45 (0.45)** | **52.81 (1.21)** | **50.70 (0.64)** |
| | GPT-4.1-nano | 22.61 (1.31) | 24.22 (1.52) | 29.63 (1.40) | 26.18 (0.89) | 30.14 (0.26) | 26.52 (1.62) | 27.03 (1.05) |
| | Gemini 2.5 Flash | **60.34 (6.32)** | 33.08 (0.77) | **56.26 (1.47)** | 25.71 (2.21) | 40.10 (0.13) | 50.19 (0.38) | 26.63 (0.06) |
| | Gemini 2.0 Flash | 44.91 (1.81) | 32.44 (0.68) | 54.34 (2.11) | 25.42 (0.51) | 36.97 (0.45) | 45.72 (1.40) | 26.12 (1.72) |
| Open | Llama 3.3-70B | 36.06 (2.03) | 35.89 (3.18) | 48.23 (1.04) | 45.04 (2.61) | 41.85 (2.72) | 43.51 (1.51) | 49.00 (2.47) |
| | Llama 3.1-8B | 26.91 (1.12) | 29.89 (1.56) | 34.70 (1.75) | 33.93 (1.76) | 31.63 (2.17) | 32.91 (0.51) | 35.59 (1.09) |
| | Mixtral-8x7B | 42.19 (1.76) | 31.84 (0.78) | 46.47 (1.75) | 32.31 (1.09) | 35.51 (0.19) | 41.24 (2.90) | 34.18 (0.96) |
| | Mistral-7B | 32.93 (0.59) | 28.20 (1.17) | 35.09 (1.76) | 30.16 (1.82) | 30.12 (1.45) | 31.00 (1.93) | 30.80 (1.75) |
| | Qwen 3-32B | 36.10 (1.16) | 33.59 (1.16) | 51.72 (2.68) | 44.13 (0.70) | 38.57 (0.21) | 45.72 (0.91) | 45.00 (0.30) |
| | Qwen 3-8B | 28.86 (1.64) | 29.25 (1.28) | 39.72 (1.13) | 34.61 (1.91) | 33.63 (1.04) | 37.59 (0.95) | 35.25 (1.61) |
| | Phi 4-14B | 24.18 (0.84) | 27.12 (1.84) | 37.93 (2.63) | 29.93 (1.24) | 30.61 (1.68) | 31.46 (1.26) | 33.63 (1.48) |
| | Phi 4 mini-3.8B | 10.34 (0.81) | 25.71 (1.22) | 30.18 (1.24) | 20.99 (0.57) | 30.99 (2.83) | 24.69 (0.87) | 21.16 (0.06) |
| | Random Guessing | 20.00 | 20.00 | 20.00 | 20.00 | 20.00 | 20.00 | 20.00 |

Table 14: The performance of the agents on the adversarial version of Friends dialogue in DialSim. We conducted experiments three times and reported the accuracy and standard deviations. **Bold** indicates the best performance per retrieval method.

| Type | Model | Base LLM | RAG-based | | | | | |
| | | | BM25 | | | OpenAI Embedding | | |
| | | | Utterance | Session Entire | Session Sum. | Utterance | Session Entire | Session Sum. |
|---|---|---|---|---|---|---|---|---|
| API | GPT-4o-mini | 39.72 (0.75) | **33.84 (0.77)** | 45.08 (1.92) | 37.36 (1.09) | **36.61 (2.14)** | 41.38 (0.54) | **42.10 (1.04)** |
| | GPT-4.1-nano | 23.56 (1.34) | 25.22 (0.70) | 27.20 (0.51) | 24.07 (0.70) | 24.78 (1.18) | 27.54 (1.99) | 24.90 (0.68) |
| | Gemini 2.5 Flash | **58.81 (3.90)** | 28.57 (0.42) | **51.40 (0.57)** | 16.77 (9.72) | 34.99 (0.77) | **43.23 (0.19)** | 24.78 (0.64) |
| | Gemini 2.0 Flash | 36.74 (3.66) | 28.14 (0.97) | 47.81 (0.39) | 23.54 (1.10) | 33.80 (0.91) | 38.53 (0.81) | 22.09 (0.75) |
| Open | Llama 3.3-70B | 31.38 (1.01) | 29.50 (1.13) | 42.87 (1.12) | **37.85 (1.22)** | 35.67 (0.57) | 38.61 (2.28) | 39.85 (0.63) |
| | Llama 3.1-8B | 26.99 (2.56) | 27.25 (1.87) | 38.48 (2.41) | 30.65 (1.01) | 29.03 (1.34) | 38.57 (0.81) | 32.99 (1.42) |
| | Mixtral-8x7B | 34.19 (0.68) | 25.23 (1.19) | 37.72 (0.96) | 29.48 (0.87) | 24.27 (1.40) | 31.80 (0.38) | 25.61 (1.21) |
| | Mistral-7B | 26.44 (1.68) | 25.24 (1.55) | 33.33 (1.26) | 26.31 (0.63) | 27.25 (0.59) | 29.76 (1.31) | 28.95 (1.51) |
| | Qwen 3-32B | 32.31 (0.73) | 30.86 (1.42) | 46.32 (2.40) | 36.31 (1.39) | 33.89 (0.47) | 39.04 (0.43) | 37.42 (0.38) |
| | Qwen 3-8B | 27.08 (0.65) | 27.08 (1.15) | 38.95 (0.45) | 30.95 (1.72) | 32.78 (1.68) | 34.18 (1.56) | 34.14 (2.68) |
| | Phi 4-14B | 19.41 (1.01) | 26.35 (0.47) | 33.04 (1.20) | 26.18 (1.18) | 27.29 (0.59) | 29.63 (0.99) | 31.08 (2.21) |
| | Phi 4 mini-3.8B | 8.98 (0.42) | 26.01 (0.43) | 30.82 (0.34) | 21.75 (0.16) | 31.59 (0.69) | 26.14 (2.07) | 23.33 (0.69) |
| | Random Guessing | 20.00 | 20.00 | 20.00 | 20.00 | 20.00 | 20.00 | 20.00 |

