# OpenReview forum: "DialSim: A Dialogue Simulator for Evaluating Long-Term Multi-Party Dialogue Understanding of Conversational Agents"
_ICLR.cc/2026/Conference — Submitted to ICLR 2026_

### Official Review · Reviewer_CSQc · 2025-10-31

**Soundness:** 3
**Presentation:** 2
**Contribution:** 2
**Rating:** 2
**Confidence:** 4

**Summary:**

In this paper, the authors propose DialSim, a simulator designed to evaluate the long-term conversation abilities of LLMs using TV series scripts. To support this framework, the authors introduce LongDialQA, a dataset constructed from refined TV series scripts that enables realistic multi-party dialogue simulation. LongDialQA anonymizes or swaps character names to prevent LLMs from relying on any prior knowledge of the shows, ensuring that models respond solely based on the given dialogue context. The dataset combines fan created (FunTrivia) and model generated questions, allowing for a diverse and comprehensive evaluation of an LLM's contextual understanding and reasoning capabilities.

**Strengths:**

1. The authors propose a simulation framework for evaluating multi-party, long-term conversation scenarios. The framework explicitly incorporates multi-hop reasoning and uncertainty handling, aiming to simulate more complex and realistic real-world conversations.
2. In building LongDialQA based on TV series scripts, the authors actively curated and refined the dataset to ensure high quality. To enhance diversity, they combined temporal knowledge graphs with fan created quizzes, creating a dataset that captures a broad range of question types, difficulty levels, and reasoning patterns.
3. The authors evaluated various open-source and proprietary LLMs on DialSim tasks. Their results show that even models with large context windows struggle to retain and reason over long-term conversational context, revealing the continuing challenges of maintaining coherent understanding across extended conversations.

**Weaknesses:**

1. Using TV show scripts to evaluate LLMs' dialogue modeling capabilities is not particularly novel. For example, MARS-Bench (Yang et al.) already employs sports commentary scripts to comprehensively assess LLM performance across various long-term conversation tasks. Considering other recent benchmarks such as LongMemEval, simply introducing another benchmark dataset of this type may not constitute a sufficiently strong contribution on its own.
2. LongDialQA primarily consists of multiple-choice questions, which may inadvertently provide hints that make the task easier for models, as LLMs can exploit option cues rather than generating answers independently. I believe the dataset should also include open-ended questions to better evaluate generative comprehension. It would also be useful to test whether results remain consistent if the order of answer choices is randomized.
3. The authors' main conclusion that increasing the context window alone is insufficient for handling long-term dialogue is somewhat simplistic. While the finding itself is valid, the paper would benefit from a deeper analysis of why this occurs. Including some analysis would make the contribution more insightful and theoretically grounded.
4. Although the paper presents DialSim and LongDialQA as its main contributions, it lacks rich and illustrative examples to help readers understand how the simulation operates in practice. Providing detailed qualitative analyses would make the framework's contribution more persuasive.

**Questions:**

1. Why did the authors restrict the evaluation to multiple-choice questions only? Would it be possible to expand the evaluation to include other answer formats or more open-ended settings?
2. LongDialQA dataset is primarily based on TV series scripts. I'm concerned that the domain specificity of TV shows might limit its effectiveness in evaluating the generalization ability of language models.
3. The authors appear to evaluate models only in a zero-shot setting. I wonder why the experiments were limited to zero-shot scenarios. It might be insightful to explore whether different prompting strategies could improve performance.
4. It seems that the questions could be categorized into multiple task types. What do the authors think about this? If possible, could you provide statistics or an analysis of the distribution of question types?
5. Please refer to weaknesses for additional feedback.

---

> ### Author Response · Authors · 2025-11-21
> **Response to Reviewer CSQc (1/5)**
>
> ```W1. Using TV show scripts to evaluate LLMs' dialogue modeling capabilities is not particularly novel. For example, MARS-Bench (Yang et al.) already employs sports commentary scripts to comprehensively assess LLM performance across various long-term conversation tasks. Considering other recent benchmarks such as LongMemEval, simply introducing another benchmark dataset of this type may not constitute a sufficiently strong contribution on its own.```
>
> We thank the reviewer for this thoughtful comment and agree that simply constructing a dataset from TV show scripts is not, in itself, a novel contribution. Rather, our main contribution lies in leveraging the long-term, multi-party nature of these scripts to more effectively evaluate a conversational agent’s dialogue understanding ability.
>
> Although MARS-Bench, like DialSim, also draws on TV show scripts, its evaluation focus is fundamentally different from ours. MARS-Bench assesses LLM capabilities in multi-turn settings through multiple tasks such as instruction following and context retrieval, whereas DialSim is a QA-style benchmark specifically designed to measure a conversational agent’s long-term, multi-party dialogue understanding. Consequently, MARS-Bench is restricted to two-party dialogue, and its conversations have an average length of 33 turns.
>
> Similarly, as we discuss in the Related Works section, LongMemEval primarily focuses on LLM-generated dialogues in a simulated user–AI assistant setting. To the best of our knowledge, there is currently no QA dataset that, like ours, combines natural multi-party dialogues with lengths of up to 350k tokens. We therefore view our dataset as complementary to prior work, as it explicitly targets this under-explored evaluation setting.

---

> ### Author Response · Authors · 2025-11-21
> **Response to Reviewer CSQc (2/5)**
>
> ```W2. LongDialQA primarily consists of multiple-choice questions, which may inadvertently provide hints that make the task easier for models, as LLMs can exploit option cues rather than generating answers independently. I believe the dataset should also include open-ended questions to better evaluate generative comprehension. It would also be useful to test whether results remain consistent if the order of answer choices is randomized.```
>
> ```Q1. Why did the authors restrict the evaluation to multiple-choice questions only? Would it be possible to expand the evaluation to include other answer formats or more open-ended settings?```
>
> As noted in the caption of Figure 1 and in Section 5.1, LongDialQA supports an open-ended question format. Details on how we evaluate in the open-ended setting are provided in Appendix G, which we summarize below.
>
> LongDialQA includes pairs of questions, answers, and choices. The questions are available in three formats: template-based multiple-choice, natural language multiple-choice, and open-ended. Users can choose any of these formats to evaluate the agent’s performance.
>
> First, we provide multiple-choice questions in both template and natural language formats. For example, a template-based question might be, “Who was going out with Paul in September 1994?” with choices “(A) Emily, (B) Monica, (C) Ryan, (D) Rachel, (E) I don’t know”. In contrast, the same question in natural language format could be phrased as, “Who was going out with Paul in September 1994? Was it Emily, Monica, Ryan, Rachel, or do you not know?”
>
> Additionally, we offer the option to ask questions in an open-ended format (e.g., “Who was going out with Paul in September 1994?”) without providing answer choices. This approach allows us to evaluate the agent’s ability to generate open-ended responses. The open-ended format is particularly useful for fan quiz-based questions, where some answers may require longer responses (e.g., Question: “Why did Monica and Chandler say they were late getting to the hospital?” Correct answer: “Monica went back for her jacket”).
>
> For natural language multiple-choice and open-ended questions, a response is considered correct if it exactly matches the correct answer. If the response does not match exactly, the score is determined by comparing the response with the correct answer using a different language model (i.e., GPT-4o mini).

---

> ### Author Response · Authors · 2025-11-21
> **Response to Reviewer CSQc (3/5)**
>
> ```W3. The authors' main conclusion that increasing the context window alone is insufficient for handling long-term dialogue is somewhat simplistic. While the finding itself is valid, the paper would benefit from a deeper analysis of why this occurs. Including some analysis would make the contribution more insightful and theoretically grounded.```
>
> Thank you for the valuable comment. To analyze this point in more depth, we note that this trend closely resembles the findings reported in LongBench and LongBench v2, which evaluate long-context understanding in LLMs. In LongBench v2, even with the same 128k-token context length, models such as Qwen-2.5 7B and GLM-4-9B-Chat achieve relatively low scores, whereas models like GPT-4o and GPT-o1 obtain much higher scores due to their stronger reasoning capabilities. We will incorporate this analysis into the main paper.
>
> In addition, following suggestions from other reviewers, we conducted further analysis on how models respond to unanswerable questions—specifically, how often they reply with “I don’t know.” The table reports, in order, the accuracy for total, answerable, and unanswerable questions, as well as the proportion of “I don’t know” responses among all answers. Interestingly, accuracy on unanswerable questions is higher than on answerable ones. This occurs because the models predominantly choose “I don’t know” when uncertain; they adopt a conservative strategy rather than guessing. This tendency is even more pronounced in models with stronger reasoning abilities. High-performing API-based models, as well as strong open-source models such as Llama 3.3 70B, show higher IDK ratios compared to weaker models. Weaker models hallucinate more often and therefore choose answers other than “I don’t know,” whereas models with stronger reasoning abilities can avoid hallucinations and explicitly admit when they do not know.
> Furthermore, even when two models achieve the same accuracy, a model like GPT-4.1-nano exhibits a much higher IDK ratio, suggesting that it may be more reliable in practical conversational settings than other models with comparable accuracy. We will add the corresponding discussion to the main paper.
>
> | Model          | Total Accuracy | Answerable | Unanswerable | IDK ratio |
> |----------------|---------------:|-----------:|-------------:|----------:|
> | gpt-4o-mini    |          52.11 |      46.24 |        75.32 |     40.87 |
> | gpt-4.1-nano   |          34.23 |      14.98 |        94.71 |     86.21 |
> | gemini-2.5-flash |        54.15 |      43.58 |        92.86 |     58.11 |
> | gemini-2.0-flash |        49.17 |      38.60 |        91.67 |     62.07 |
> | llama-3.3-70b-it |        48.15 |      37.94 |        87.58 |     57.22 |
> | llama-3.1-8b-it |         39.85 |      29.19 |        80.37 |     50.45 |
> | mixtral-it     |          40.36 |      38.17 |        49.66 |     34.48 |
> | mistral-7b-it  |          39.21 |      34.45 |        58.33 |     46.23 |
> | qwen-3-8b      |          41.89 |      37.44 |        59.49 |     36.65 |
> | qwen-3-32b     |          46.23 |      37.66 |        79.87 |     49.81 |
> | phi-4          |          31.16 |      22.64 |        68.03 |     38.44 |
> | phi-4-mini     |          33.59 |      29.05 |        48.37 |     48.15 |

---

> ### Author Response · Authors · 2025-11-21
> **Response to Reviewer CSQc (4/5)**
>
> ```W4. Although the paper presents DialSim and LongDialQA as its main contributions, it lacks rich and illustrative examples to help readers understand how the simulation operates in practice. Providing detailed qualitative analyses would make the framework's contribution more persuasive.```
>
> Thank you for the valuable feedback. We will add these examples on the revised manuscript.
>
>
> ###########################################
> Date: May 23, 1999, Conversation #1
> ###########################################
>
> Linda:  Okay umm, Robert? I'm-I'm really warm, so I'm going to be taking off my sweater. Now, I'm just letting you know that this is not an invitation to the physical act of love.
> Robert:  Yep! That's hilarious!
> Linda: I'm sorry. I'm done. I'm done.
> Robert: Y'know, last night was embarrassing for you too.
> Linda: No, not really. I mean you've seen me naked hundreds of times.
> Robert: Uh-huh. But it was a first for the rest of my building.
> Linda: Okay. All right, that's true! But y'know I just don't embarrass that easily.
> Robert: What?! You totally get embarrassed!
> Linda: No, I don't! Robert, I think I'm just a more secure person than you are.
> Robert: Is that so?
> Linda: Oh. My. God. Remember when Linda had a roommate back in April 1998? So, who was going out with that roommate by September 1994?
>         (A) Paul
>         (B) Robert
>         (C) Katie
>         (D) Robert
>         (E) I don't know. (None of the above)
> ------------------------------- Q&A result -------------------------------
> result: Wrong, ambiguous answer: False
> true answer: (A)         model answer: (E) I don't know. (None of the above)
> model name: gpt-4o-mini
> ret_method: bm25
> history_type: utts
>
> Linda: Yeah.
> Robert:  Hey lady! I don't care how much you want it! Okay?! I am not gonna to have sex with you in the bathroom!
>
> ###########################################
> Date: May 23, 1999, Conversation #2
> ###########################################
>
> Guy:  So uh, I'm on my way back to the bathroom.
> Linda: Yeah, all right. All right! Just keep walkin'! All right?
> Linda: Robert! What are you… I'm sorry sir. I just, I think he just really likes you.
> Robert:  Hey! Y'know that teacher who had a baby with her student?
> Robert: What the? What…
> Linda: Oh my God, did you hear about why Robert suddenly changed his plans and hung out in the city for an extra day?
>         (A) A Van Gogh exhibit
>         (B) A paleontologist tour he was giving
>         (C) Ben was in a play
>         (D) He did not want Linda to have to fly alone
>         (E) I don't know. (None of the above)
> ------------------------------- Q&A result -------------------------------
> result: Wrong, ambiguous answer: False
> true answer: (A)         model answer: (D) He did not want Linda to have to fly alone
> model name: gpt-4o-mini
> ret_method: bm25
>
> history_type: utts
>
> Linda:  Hi!
> The Flight Attendant: Miss? May I help you?
> Linda: Yes, I'm sorry. Do you have any extra pants? Umm, my friend seems to have had a little accident.

---

> ### Author Response · Authors · 2025-11-21
> **Response to Reviewer CSQc (5/5)**
>
> ```Q2. LongDialQA dataset is primarily based on TV series scripts. I'm concerned that the domain specificity of TV shows might limit its effectiveness in evaluating the generalization ability of language models.```
>
> Thank you for the valuable question. We agree that reliance on TV shows raises concerns about domain generalizability, and we intentionally designed our setup to partially mitigate this. The three series already cover a range of conversational settings—interpersonal relationships (Friends), academic discourse (The Big Bang Theory), and workplace interactions (The Office).
>
> We initially explored constructing DialSim from real-world multi-party dialogue corpora as well, since we agree that this would further strengthen its applicability. However, in practice we encountered substantial limitations: many promising sources (e.g., customer service or retail interactions) are proprietary corporate assets, subject to strict privacy constraints, and therefore difficult to obtain at scale. Given these constraints, we opted for high-quality TV show scripts as a feasible proxy, while designing our question-generation framework so that it can be readily applied to future real-world dialogue datasets once they become available.
>
>
> ```Q3. The authors appear to evaluate models only in a zero-shot setting. I wonder why the experiments were limited to zero-shot scenarios. It might be insightful to explore whether different prompting strategies could improve performance.```
>
> Thank you for this interesting question. Our evaluation focuses on a zero-shot setting that is intended to approximate a realistic scenario in which a conversational agent participates in an ongoing multi-party dialogue. In such scenarios, the agent cannot anticipate when it will be addressed or what kinds of questions it will receive. Thus, assuming access to question–answer pairs that are similar to an incoming query, or inserting carefully curated QA examples as in-context demonstrations, presupposes prior knowledge about the queries and does not align with the nature of real conversational interactions. For this reason, we believe that zero-shot evaluation best reflects the intended use case, and we therefore conducted our experiments in a zero-shot setting. We will clarify this point in the revised manuscript.
>
> ```Q4. It seems that the questions could be categorized into multiple task types. What do the authors think about this? If possible, could you provide statistics or an analysis of the distribution of question types?```
>
> Could you explain the notion of multiple task types/question types in more detail? If the concern refers to an analysis of question types, we address this in Section 5.3. Specifically, we analyzed the results of the Gemini-2.5 Flash Base LLM, as it achieved the highest overall performance. The results show that fan-quiz-based questions achieved an accuracy of 62.15%, while TKG-based questions performed lower at 50.83%, indicating that TKG-based questions are more challenging. When we further break down the TKG-based questions, one-hop questions achieved 69.19%, whereas two-hop questions scored only 19.28%, highlighting the significant difficulty posed by two-hop questions.

---

> > ### Comment · Reviewer_CSQc · 2025-11-23
> >
> > **Thank you to the authors for their responses to my concerns and questions.**
> >
> >
> > **Weaknesses 2 and Questions 1**
> >
> > As the authors said, we can find the experimental setup in Section 5.1. I know this information too. However, as the authors wrote in line 359-360, this work uses only the multiple-choice format. I also could not find any examples or results for open-ended questions in Appendix G.
> >
> > What I want to see, exactly as the authors described, is the model's performance on open-ended questions without any answer choices. The authors say that they used multiple-choice questions for efficiency, but I think showing at least some open-ended results would make the paper stronger and more insightful. Just saying that this work supports open-ended questions does not fully answer my concern. If I am misunderstanding something, please let me know.
> >
> > **Weaknesses 1 and Weaknesses 3**
> >
> > I agree with the authors that one of the main contributions of this paper is the "multi-party" setting. I also read the further analysis suggested by the other reviewers, but I still think more analysis is needed.
> >
> > For example, long-term reasoning on dyadic conversations has already been studied. Since this paper claims a contribution in the multi-party setting, I think it would be useful to analyze what becomes harder specifically because the conversation has many speakers. For example, analysis about the context window is helpful, but this kind of analysis can be done even without multi-party data.
> >
> > **Weaknesses 4 and Questions 2**
> >
> > Thank you for the authors' answer. I hope these details will be included in the revised version.
> >
> > **Questions 3**
> >
> > I understand why the evaluation uses a zero-shot setting, and I agree that this is the appropriate choice for real-world conversational scenarios. However, I was curious if using methods like CoT or prompting could improve the results. Usually, when zero-shot performance is not strong, people try these methods. So I wonder if they would help in this case too.
> >
> >
> > **Questions 4**
> >
> > What I wanted to know was something like this: for example, does the model perform worse when time information is included compared to when it is not included? The results the authors shared are also interesting.
> >
> >
> > **I hope the authors will address my remaining concerns during the remaining rebuttal period. Therefore, I will keep my score unchanged for now.**

---

> > > ### Author Response · Authors · 2025-12-03
> > > **Final Response to Reviewer CSQc (1/2)**
> > >
> > > **Weaknesse 2 and Questions 1**
> > >
> > > Thank you for the valuable feedback. The performance of each conversational agent under the open-ended setting is summarized in the table below. Compared to the multiple-choice format, the scores dropped substantially; in particular, even the best-performing configuration—Gemini 2.5 Flash with OpenAI Embedding-Session Entire—achieved only 32.74%. This indicates that providing five answer options in the multiple-choice format served as a strong hint, and the very low performance in the open-ended format directly reveals the current limitations of conversational agents in long-term multi-party dialogue understanding. We will add a discussion about this point in the revised manuscript.
> > >
> > > | Type      | Model             | Base LLM |        |    BM25   |  |        |     OpenAI Embedding         |  |        |               |
> > > |-----------|-------------------|--------------|--------|--------|------------------|--------|---------------|-------------------------------|--------|---------------|
> > > |           |                   |     | Utt              | Session Entire | Session Sum. | Utt                           | Session Entire | Session Sum. |
> > > | API based  | gemini 2.5 flash| 29.54                    | 21.55    | 26.01                | 23.42             | 28.6                 | 32.74                            | 30.54                          |
> > > | API based  | gpt-4o-mini     | 28.86                    | 22.73    | 25.93                | 19.92             | 23.37                | 25.03                            | 24.65                          |
> > > | Opensource  | Llama 3.3 70b   | 25.03                    | 17.50    | 23.75                | 22.61             | 18.29                | 20.05                            | 18.01                          |
> > > | Opensource  | Llama 3.1 8b    | 19.80                    | 18.52    | 21.46                | 18.52             | 15.28                | 18.41                            | 16.35                          |
> > > | Opensource  | phi-4-mini      | 10.34                    | 13.79    | 12.64                | 5.24              | 9.25                 | 9.32                             | 9.58                           |
> > > | Opensource | qwen-3-8b       | 21.32                    | 24.36    | 25.89                | 20.58             | 22.76                | 23.60                            | 20.72                          |
> > >
> > >
> > >
> > > **Weaknesses 1 and Weaknesses 3**
> > >
> > > Thank you for your thoughtful feedback. To investigate how the number of participants in a conversation affects dialogue understanding, we measured performance in the Gemini 2.5 Flash oracle setting based on the total number of unique speakers appearing in the evidence sessions. We found that accuracy drops sharply as the number of speakers increases: while accuracy is 82.20% when there are 1–5 speakers, it falls to 12.50% in the 20–25 speaker range. This indicates that dialogue reasoning becomes significantly more challenging as the number of participants increases—a finding that cannot be observed in two-party dialogue QA. We will include a discussion of this result in the revised manuscript.
> > >
> > > | # of Characters | Accuracy |
> > > |---------|-------------|
> > > | 1-5       | 82.20%        |
> > > | 6-10       | 34.98%        |
> > > | 11-15       | 20.00%        |
> > > | 16-20       | 16.83%        |
> > > | 21-15       | 12.50%        |

---

> > > ### Author Response · Authors · 2025-12-03
> > > **Final Response to Reviewer CSQc (2/2)**
> > >
> > > **Questions 3**
> > >
> > > Thank you for the valuable feedback. The original prompt involved selecting just one choice from multiple choices for evaluation, without additional reasoning. However, as you suggested, we conducted further experiments using a CoT prompt, and the evaluation was carried out with GPT-4o-mini using LLM-as-a-Judge. As a result, the findings are shown in the table below. In most cases, there was a performance improvement of around 2% to 10%, and notably, in the Gemini 2.5 flash-Base LLM setting, it achieved a score of 76.21%, which is similar to the oracle setting's performance. It was an intriguing observation to see such significant performance improvement with just a change in the prompting method, and we will include a discussion of this in the revised manuscript.
> > >
> > > | Type      | Model             | Base LLM |        |    BM25   |  |        |     OpenAI Embedding         |  |        |               |
> > > |-----------|-------------------|--------------|--------|--------|------------------|--------|---------------|-------------------------------|--------|---------------|
> > > |           |                   |     | Utt              | Session Entire | Session Sum. | Utt                           | Session Entire | Session Sum. |
> > > | API based | gemini 2.5 flash  | 76.21        | 29.89  |         54.79           | 34.69         |                35.12                        | 44.32          | 34.41        |
> > > | API based | gpt-4o-mini       | 52.94        | 31.03  |        45.98           | 34.99         |                39.00                           | 41.51          | 41.12        |
> > > | Opensource | Llama 3.1 8b      | 40.64        | 39.13  |        47.82           | 42.04         |                43.00                           | 45.96          | 46.33        |
> > > | Opensource | phi-4             | 34.96        | 32.24  |         49.01           | 40.75         |                37.35                        | 30.40           | 39.15        |
> > > | Opensource | phi-4-mini        | 33.44        | 28.65  |        34.66           | 28.38         |                35.13                        | 35.22          | 29.56        |
> > >
> > >
> > > **Questions 4**
> > >
> > > Thank you for your thoughtful feedback. As you suggested, we conducted an additional analysis on the impact of time information on model performance. Using the results of gpt-4o-mini BM25-Session Entire, which showed the best performance, we compared questions without time information and questions with time information. The accuracy for questions without time information was 76.23%, whereas the accuracy for questions with time information was 36.73%. This appears to be due to the relatively large number of unanswerable questions—such as those asking about events that occur in the future—being included in the “without temporal information” set, which inflates the performance.
> > >
> > > We also compared performance within the time-included questions by separating those that contained only a year or year-month from those that contained a full date (year-month-day). The accuracies were 30.26% and 39.76%, respectively. We interpret this as evidence that retrieval performs better when the time information is more specific (e.g., a full date), compared to broader time expressions such as only a year or a year and month.
> > >
> > > We will include a discussion of these findings in the revised manuscript.

---

### Official Review · Reviewer_Kgeb · 2025-10-31

**Soundness:** 3
**Presentation:** 4
**Contribution:** 4
**Rating:** 6
**Confidence:** 4

**Summary:**

This paper introduces DialSim, which is a new way to test if chatbots can understand long conversations between many people. The authors made a dataset to support this, called LongDialQA, which contains dialogs from TV shows like Friends, The Big Bang Theory, and The Office. For every chat, there are different types of questions, some of which are straightforward, some needs memory and reasoning, and some that cannot be answered from the chat context. In DiamSim, during testing, the chatbot takes the role of one character and must answer questions using only the dialogue given. If it doesn’t have enough info, it should say, "I don’t know". The authors tested some language models and the results were not great, even for larger models. None of them scored more than 60%, especially on questions that needed remembering older parts of the chat or connecting many points together.

**Strengths:**

- The paper's focus on long, multi-party chats is a setup that is needed and is usable in real-world evaluation.
- I like the inclusion of questions that are not answerable. Making the models learn when to say "I don't know" is a separate, a very interesting, research direction in itself.
- The paper is very well written and is easy to follow.

**Weaknesses:**

- Although the authors anonymize the chats by changing or removing speaker names from dialogues, the models could still have prior knowledge about the sitcoms by analyzing the context of the chat. Moreover, the use of TV show scripts does not reflect real world conversations well, which are much more messy. Although DialSim is an interesting evalaution setup, testing it on other real-world data (like meeting notes, etc) could say much more about its efficiency.
- The oracle test shows higher scores than RAG, but still fails on many questions. This means the real problem is not just finding the right info, but it is the process of reasoning over time, which the paper does not highlight enough.
- The paper can include some statistics about how many time does the model answer "I don't know" to see if it is using it too often or too little.

**Questions:**

- In the oracle setting, the two-hop questions still have ~48% error. Why could be that? Maybe due to temporal reasoning failures, entity linking errors, or something else?
- Did you observe any systematic pattern in "I don’t know" usage? For example, do smaller models default to it more often?

---

> ### Author Response · Authors · 2025-11-21
> **Response to Reviewer Kgeb (1/4)**
>
> ```W1. Although the authors anonymize the chats by changing or removing speaker names from dialogues, the models could still have prior knowledge about the sitcoms by analyzing the context of the chat.```
>
> Thank you for your thoughtful feedback. We acknowledge the limitation that many LLMs may have been exposed to these scripts during pretraining, meaning prior knowledge could influence performance. To mitigate this, we applied two strategies: character name anonymization and name swapping. In additional experiments using a recent strong-reasoning model (Gemini-2.5 Pro), we found that anonymization alone was sometimes insufficient: in several QA cases, the model was able to infer the original names and use memorized knowledge to answer the questions. This indicates that anonymizing character names does not fully eliminate prior-knowledge effects.
>
> Name swapping, however, is used in a more adversarial way. We construct questions so that any reliance on memorized character facts would increase the chance of a wrong answer, forcing the model to depend only on the dialogue content. From this perspective, we believe name swapping provides a stronger adversarial evaluation and better matches the purpose of stress-testing conversational understanding.
>
> If we have misunderstood any aspect of your concern, please let us know—we would be very happy to revise accordingly.

---

> ### Author Response · Authors · 2025-11-21
> **Response to Reviewer Kgeb (2/4)**
>
> ```W2. Moreover, the use of TV show scripts does not reflect real world conversations well, which are much more messy. Although DialSim is an interesting evalaution setup, testing it on other real-world data (like meeting notes, etc) could say much more about its efficiency.```
>
> Thank you for your valuable feedback. We agree that reliance on TV shows raises concerns about domain generalizability, and we intentionally designed our setup to partially mitigate this. The three series already cover a range of conversational settings—interpersonal relationships (Friends), academic discourse (The Big Bang Theory), and workplace interactions (The Office).
>
> We initially explored constructing DialSim from real-world multi-party dialogue corpora as well, since we agree that this would further strengthen its applicability. However, in practice we encountered substantial limitations: many promising sources (e.g., meeting notes or customer services) are proprietary corporate assets, subject to strict privacy constraints, and therefore difficult to obtain at scale. Given these constraints, we opted for high-quality TV show scripts as a feasible proxy, while designing our question-generation framework so that it can be readily applied to future real-world dialogue datasets once they become available.

---

> ### Author Response · Authors · 2025-11-21
> **Response to Reviewer Kgeb (3/4)**
>
> ```W3. The oracle test shows higher scores than RAG, but still fails on many questions. This means the real problem is not just finding the right info, but it is the process of reasoning over time, which the paper does not highlight enough.```
>
> ```Q1. In the oracle setting, the two-hop questions still have ~48% error. Why could be that? Maybe due to temporal reasoning failures, entity linking errors, or something else?```
>
> Thank you for this very helpful feedback. In the oracle setting, evidence sessions are provided based on their dates. For two-hop questions, the dialogue sessions corresponding to the two dates can be as long as 8k tokens. Accurately locating the answer to a two-hop question within such long dialogues can be challenging for models. This is because the answer may not appear explicitly in the dialogue but may instead be expressed implicitly.
>
> For example, consider a question like: “Who was Robert (Ross)’s sister’s boyfriend in 1995?”  To answer this, the model must first infer who Robert’s sister is, using information that is only implicitly revealed across the dialogue sessions. Then, the model must also infer who her boyfriend was by understanding the conversational context. Since such information is distributed across long dialogues and requires multi-hop inference, this task is challenging not only for information retrieval but also for dialogue reasoning.

---

> ### Author Response · Authors · 2025-11-21
> **Response to Reviewer Kgeb (4/4)**
>
> ```W4. The paper can include some statistics about how many time does the model answer "I don't know" to see if it is using it too often or too little.```
>
> ```Q2. Did you observe any systematic pattern in "I don’t know" usage? For example, do smaller models default to it more often?```
>
> Thank you for the valuable question. The table shows, in order, the accuracy for the total, answerable, and unanswerable questions, as well as the proportion of “I don’t know” responses among all answers. Interestingly, the accuracy on unanswerable questions is higher than the accuracy on answerable ones. This is because the models predominantly choose “I don’t know” more often than any other choice; when they are uncertain about an answer, they tend to respond conservatively by selecting “I don’t know.” This tendency is even more pronounced in models with stronger reasoning ability. API-based models that perform well on other benchmarks, as well as high-performing open-source models such as Llama 3.3 70B, show higher IDK ratios compared to weaker models. Weaker models hallucinate more easily and therefore choose answers other than “I don’t know,” whereas models with stronger reasoning abilities can avoid hallucinations and admit when they do not know. Furthermore, even when accuracy is the same, a model like GPT-4.1-nano shows a much higher IDK ratio, suggesting that in practical conversational settings it may be more reliable than other models with similar accuracy. We will add a corresponding discussion to the main paper.
>
>
> | Model          | Total Accuracy | Answerable | Unanswerable | IDK ratio |
> |----------------|---------------:|-----------:|-------------:|----------:|
> | gpt-4o-mini    |          52.11 |      46.24 |        75.32 |     40.87 |
> | gpt-4.1-nano   |          34.23 |      14.98 |        94.71 |     86.21 |
> | gemini-2.5-flash |        54.15 |      43.58 |        92.86 |     58.11 |
> | gemini-2.0-flash |        49.17 |      38.60 |        91.67 |     62.07 |
> | llama-3.3-70b-it |        48.15 |      37.94 |        87.58 |     57.22 |
> | llama-3.1-8b-it |         39.85 |      29.19 |        80.37 |     50.45 |
> | mixtral-it     |          40.36 |      38.17 |        49.66 |     34.48 |
> | mistral-7b-it  |          39.21 |      34.45 |        58.33 |     46.23 |
> | qwen-3-8b      |          41.89 |      37.44 |        59.49 |     36.65 |
> | qwen-3-32b     |          46.23 |      37.66 |        79.87 |     49.81 |
> | phi-4          |          31.16 |      22.64 |        68.03 |     38.44 |
> | phi-4-mini     |          33.59 |      29.05 |        48.37 |     48.15 |

---

### Official Review · Reviewer_yeJG · 2025-11-04

**Soundness:** 2
**Presentation:** 2
**Contribution:** 2
**Rating:** 2
**Confidence:** 4

**Summary:**

The paper introduces DialSim, a simulation framework for evaluating conversational agents’ ability to track and reason over long-term, multi-party dialogues. In DialSim, an agent “plays” a named character inside a scripted conversation and must answer spontaneous questions only from the dialogue history, selecting “I don’t know” when appropriate. To power the simulator, the authors construct LongDialQA from multi-season TV scripts (Friends, The Big Bang Theory, The Office): ~1,300 sessions with >1k questions per session and dialogue lengths on the order of ~300–350k tokens per show; character names are anonymized and an adversarial variant swaps names to reduce prior-knowledge leakage.

**Strengths:**

1. Realistic long-horizon, multi-party setup with abstention. The simulator blends long context, multiple speakers, and uncertainty (“I don’t know”)
2. Careful dataset construction and controls. The combination of fan-quiz evidence mapping, TKG-driven multi-hop questions, character style transfer, and anonymization/adversarial name swapping is thoughtful

**Weaknesses:**

1. Retrieval baselines feel narrow. RAG uses BM25 and vendor embeddings; no dense retrievers tuned for dialogue, no temporal/speaker-aware indexing, and no hierarchical retrieval. Given the oracle gap (+10–30%), richer retrievers could materially change conclusions.
Few options to consider: Contriever (unsupervised dense IR), ColBERTv2 (late-interaction), and E5 family (text-embedding models for retrieval).

2. Question timing and asker selection may distort conversational realism. The scheduler injects questions at random positions and chooses askers within three turns of the agent’s last utterance; this can yield unnatural density or speaker distributions. Consider reporting statistics on injection timing and a control where questions are asked only in plausible discourse slots.


3. Construction relies heavily on GPT-4 with limited quality metrics. GPT-4 is used to filter quiz questions, map evidence scenes, extract triples, and style-transfer; the paper states “authors verified,” but provides no inter-annotator agreement, adjudication protocol, or noise estimates.

**Questions:**

See weakness.

---

> ### Author Response · Authors · 2025-11-21
> **Response to Reviewer yeJG (1/3)**
>
> ```W1. Retrieval baselines feel narrow. RAG uses BM25 and vendor embeddings; no dense retrievers tuned for dialogue, no temporal/speaker-aware indexing, and no hierarchical retrieval. Given the oracle gap (+10–30%), richer retrievers could materially change conclusions. Few options to consider: Contriever (unsupervised dense IR), ColBERTv2 (late-interaction), and E5 family (text-embedding models for retrieval).```
>
> Thank you for the helpful feedback. Our primary goal in this work is to build a benchmark for evaluating conversational agents, rather than to design an agent that is highly optimized for this specific benchmark. For this reason, we focused on implementing agents that can operate in long-term dialogue settings using representative, widely used retrieval methods and standard history-storage strategies, and we evaluated their capabilities in that setting.
>
> In response to your suggestion, we additionally constructed conversational agents that use more advanced retrieval models, including Contriever and E5-base-v2, for some of the better-performing LLMs, and evaluated them on DialSim. The corresponding results are reported in the following table. Interestingly, even with these stronger retrievers, we observed only modest performance gains over BM25 and the vendor-provided embeddings. This suggests a limitation of the current paradigm of simply retrieving highly relevant past dialogue segments and feeding them to the agent. As you point out, more substantial improvements on DialSim likely require retrieval architectures explicitly tailored to dialogue—such as dialogue-specific retrievers, temporal/speaker-aware indexing, or even GraphRAG-style methods that reason over entities and events mentioned in the conversation. We will add a discussion of these directions to the revised manuscript.
>
> | Type        | Model            | BM25 Utt | BM25 Session Entire | OpenAI Embedding Utt | OpenAI Embedding Session Entire | | Contriever Utt | Contriever Session Entire | E5 Utt | E5 Session Entire |
> |-------------|------------------|---------:|---------------------:|----------------------:|---------------------------------:|-:|----------------:|----------------------------:|-------:|-------------------:|
> | API based   | gemini 2.5 flash |    30.86 |               `52.92`  |                 37.16 |                            42.40 | |          30.01 |                       44.44 |   36.53 |             49.04 |
> | API based   | gpt-4o-mini      |    35.46 |               ``52.11``  |                 41.29 |                            45.47 | |          34.87 |                       44.70 |   43.68 |             51.09 |
> | Opensource  | Llama 3.3 70b    |    27.80 |               44.91  |                 38.70 |                            39.63 | |          32.44 |                       38.83 |   35.89 |             ``46.62`` |
> | Opensource  | Mistral          |    27.71 |               37.85  |                 30.01 |                            34.99 | |          28.22 |                       31.16 |   31.93 |             ``38.70`` |
> | Opensource  | qwen 3 32B       |    30.82 |               ``47.55``  |                 36.44 |                            42.87 | |          32.31 |                       42.15 |   39.34 |             43.55 |

---

> ### Author Response · Authors · 2025-11-21
> **Response to Reviewer yeJG (2/3)**
>
> ```W2. Question timing and asker selection may distort conversational realism. The scheduler injects questions at random positions and chooses askers within three turns of the agent’s last utterance; this can yield unnatural density or speaker distributions. Consider reporting statistics on injection timing and a control where questions are asked only in plausible discourse slots.```
>
> Thank you very much for the constructive feedback. Our goal was not to recreate fully naturalistic dialogues, but rather to evaluate whether a conversational agent can understand the preceding context and answer questions that appear at arbitrary moments. Nonetheless, we adopted character-style transfer and restricted the asker to speakers who appeared within the past three turns in order to avoid highly implausible question–answering scenarios. While this design may introduce some unnatural situations, all injected questions and answers remain logically grounded in, and referable to, prior session content. Therefore, we believe they still effectively test the model’s ability to understand long-term dialogue, and that these potential artifacts do not compromise the overall validity of our evaluation.
>
> Regarding the distribution of askers, the Friends dialogue yielded the following proportions: Jennifer (Phoebe): 10.97%, Patricia (Monica): 13.49%, John (Joey): 15.14%, James (Chandler): 16.20%, and Linda (Rachel): 23.42%. Although the askers are not uniformly distributed, their relative frequencies naturally correlate with each character’s share of utterances: Jennifer (Phoebe): 13.07%, John (Joey): 15.11%, Patricia (Monica): 15.62%, James (Chandler): 16.42%, and Linda (Rachel): 20.43%. As for injection timing, because the question positions are sampled randomly, the resulting statistics reflect this and appear uniformly distributed. We will include a corresponding discussion in the main paper.

---

> ### Author Response · Authors · 2025-11-21
> **Response to Reviewer yeJG (3/3)**
>
> ```W3. Construction relies heavily on GPT-4 with limited quality metrics. GPT-4 is used to filter quiz questions, map evidence scenes, extract triples, and style-transfer; the paper states “authors verified,” but provides no inter-annotator agreement, adjudication protocol, or noise estimates.```
>
> We appreciate the reviewer’s concern regarding our reliance on GPT-4 during dataset construction and the lack of explicit quality metrics. GPT-4 was used solely as a pseudo-labeling tool to assist and streamline human annotation, rather than as a fully automated labeling pipeline. For tasks such as triple extraction and evidence-scene mapping, GPT-4 was employed to propose initial candidates, but all outputs were subsequently reviewed and finalized by the authors. During this process, we removed overly verbose triples and carefully corrected or discarded incorrectly mapped evidence scenes.
>
> To further assess annotation quality for each task, an author who was not involved in the original annotation independently reviewed separate samples for quiz filtering, evidence-scene mapping, triple extraction, and style transfer. For quiz filtering, this review of 100 sampled items found 99% to be correct. For evidence-scene mapping, 97% of 100 sampled items were confirmed to be correct. For triple extraction, 94% of 50 sampled sessions were judged correct, and for style transfer, 96.5% of 200 sampled outputs were judged acceptable.
>
> We will incorporate these clarifications and the corresponding agreement statistics into the main text to make our quality-control procedures more transparent.

---

### Official Review · Reviewer_P37e · 2025-11-12

**Soundness:** 3
**Presentation:** 3
**Contribution:** 3
**Rating:** 6
**Confidence:** 4

**Summary:**

This paper introduces DialSim, a simulation-based benchmark that evaluates conversational agents’ understanding in long-term, multi-party dialogues using its newly proposed dataset, LongDialQA, derived from five seasons of three popular TV shows. Experiments show that SOTA LLMs struggle to score high on the benchmark, especially in multi-hop reasoning and long-span dependencies.

**Strengths:**

1. The paper presents an effective simulation-based evaluation paradigm for long-term multi-party dialogue understanding with specially designed questions and answer choices (with "I don't know" choice).

2. The paper provides a large-scale, multi-party, long-horizon dataset that spans five seasons across three shows, with 1300+ sessions, ~352K tokens, and 1000+ curated questions per session, surpassing prior two-party, shorter datasets. Empirical results show that all models score under 60% on the benchmark, highlighting its difficulty and value.

3. The paper has a rigorous question curation pipeline and adds temporal knowledge to multi-hop reasoning and time-conditioned queries.

4. The paper conducts comprehensive, comparative experiments to evaluate SOTA LLM APIs and open-sourced LLMs across different retrieval methods and storage granularities.

**Weaknesses:**

1. The dataset is built from three very popular entertainment-focused TV shows, which limits its generalizability across domains and applications in real-world scenarios.

2. Even with anonymization or name swapping, prior knowledge is not fully eliminated. Models can still leverage memorized show knowledge. Swapping names is not a robust and effective adversarial strategy because it may create contradictions with memorized character attributes rather than blocking access to prior facts. Consider stronger adversarial controls beyond name swaps.

**Questions:**

1. Could you report disaggregated metrics isolating answerable vs. unanswerable and leakage-sensitive subsets to quantify effect of prior knowledge?

2. Simulation-based evaluation seems to be an expensive method. Could you report the evaluation latency and cost metrics across models and settings?

---

> ### Author Response · Authors · 2025-11-21
> **Response to Reviewer P37e (1/4)**
>
> ```W1. The dataset is built from three very popular entertainment-focused TV shows, which limits its generalizability across domains and applications in real-world scenarios.```
>
> Thank you for your valuable feedback. We agree that reliance on TV shows raises concerns about domain generalizability, and we intentionally designed our setup to partially mitigate this. The three series already cover a range of conversational settings—interpersonal relationships (Friends), academic discourse (The Big Bang Theory), and workplace interactions (The Office).
>
> We initially explored constructing DialSim from real-world multi-party dialogue corpora as well, since we agree that this would further strengthen its applicability. However, in practice we encountered substantial limitations: many promising sources (e.g., customer service or retail interactions) are proprietary corporate assets, subject to strict privacy constraints, and therefore difficult to obtain at scale. Given these constraints, we opted for high-quality TV show scripts as a feasible proxy, while designing our question-generation framework so that it can be readily applied to future real-world dialogue datasets once they become available.

---

> ### Author Response · Authors · 2025-11-21
> **Response to Reviewer P37e (2/4)**
>
> ```W2. Even with anonymization or name swapping, prior knowledge is not fully eliminated. Models can still leverage memorized show knowledge. Swapping names is not a robust and effective adversarial strategy because it may create contradictions with memorized character attributes rather than blocking access to prior facts. Consider stronger adversarial controls beyond name swaps.```
>
> Thank you for your thoughtful feedback. Our original goal was to find a long-running, coherent dialogue dataset that reflects multi-year interactions between the same participants. However, we were unable to locate any publicly available datasets capturing real human conversations of this kind, and creating a new dataset with dialogues spanning five or more years was not feasible. For this reason, we chose TV show scripts, which naturally contain diverse speakers and sustained conversations over long periods.
>
> We acknowledge the limitation that many LLMs may have been exposed to these scripts during pretraining, meaning prior knowledge could influence performance. To mitigate this, we applied two strategies: character name anonymization and name swapping. In additional experiments using a recent strong-reasoning model (Gemini-2.5 Pro), we found that anonymization alone was sometimes insufficient: in several QA cases, the model was able to infer the original names and use memorized knowledge to answer the questions. This indicates that anonymizing character names does not fully eliminate prior-knowledge effects.
>
> Name swapping, however, is used in a more adversarial way. We construct questions so that any reliance on memorized character facts would increase the chance of a wrong answer, forcing the model to depend only on the dialogue content. From this perspective, we believe name swapping provides a stronger adversarial evaluation and better matches the purpose of stress-testing conversational understanding. At the same time, we agree with your point that swapping can introduce contradictions with memorized character attributes rather than completely blocking access to prior knowledge. Therefore, we interpret real-world conversational agent performance as likely lying between the results from anonymization and from name swapping.
>
> If we have misunderstood any aspect of your concern, please let us know—we would be very happy to revise accordingly.

---

> ### Author Response · Authors · 2025-11-21
> **Response to Reviewer P37e (3/4)**
>
> ```Q1. Could you report disaggregated metrics isolating answerable vs. unanswerable and leakage-sensitive subsets to quantify effect of prior knowledge?```
>
> Thank you for the valuable question. The table shows, in order, the accuracy for the total, answerable, and unanswerable questions, as well as the proportion of “I don’t know” responses among all answers. Interestingly, the accuracy on unanswerable questions is higher than the accuracy on answerable ones. This is because the models predominantly choose “I don’t know” more often than any other choice; when they are uncertain about an answer, they tend to respond conservatively by selecting “I don’t know.” This tendency is even more pronounced in models with stronger reasoning ability. API-based models that perform well on other benchmarks, as well as high-performing open-source models such as Llama 3.3 70B, show higher IDK ratios compared to weaker models. Weaker models hallucinate more easily and therefore choose answers other than “I don’t know,” whereas models with stronger reasoning abilities can avoid hallucinations and admit when they do not know. Furthermore, even when accuracy is the same, a model like GPT-4.1-nano shows a much higher IDK ratio, suggesting that in practical conversational settings it may be more reliable than other models with similar accuracy.
>
> The original questions that did not undergo character anonymization or name swapping are leakage-sensitive because an LLM that has memorized all TV show content could answer all of them correctly. To prevent this, we modified the dialogues and questions using character anonymization or name swapping. If we have misunderstood what you mean by the leakage-sensitive subset, please let us know with further clarification.
>
> | Model          | Total Accuracy | Answerable | Unanswerable | IDK ratio |
> |----------------|---------------:|-----------:|-------------:|----------:|
> | gpt-4o-mini    |          52.11 |      46.24 |        75.32 |     40.87 |
> | gpt-4.1-nano   |          34.23 |      14.98 |        94.71 |     86.21 |
> | gemini-2.5-flash |        54.15 |      43.58 |        92.86 |     58.11 |
> | gemini-2.0-flash |        49.17 |      38.60 |        91.67 |     62.07 |
> | llama-3.3-70b-it |        48.15 |      37.94 |        87.58 |     57.22 |
> | llama-3.1-8b-it |         39.85 |      29.19 |        80.37 |     50.45 |
> | mixtral-it     |          40.36 |      38.17 |        49.66 |     34.48 |
> | mistral-7b-it  |          39.21 |      34.45 |        58.33 |     46.23 |
> | qwen-3-8b      |          41.89 |      37.44 |        59.49 |     36.65 |
> | qwen-3-32b     |          46.23 |      37.66 |        79.87 |     49.81 |
> | phi-4          |          31.16 |      22.64 |        68.03 |     38.44 |
> | phi-4-mini     |          33.59 |      29.05 |        48.37 |     48.15 |

---

> ### Author Response · Authors · 2025-11-21
> **Response to Reviewer P37e (4/4)**
>
> ```Q2. Simulation-based evaluation seems to be an expensive method. Could you report the evaluation latency and cost metrics across models and settings?```
>
> Thank you for this helpful comment. We agree that simulation-based evaluation can be expensive, so we report both latency and cost-related metrics. We compute the cost per question based on the average input/output token usage and the average runtime per experiment, as summarized below.
>
> For a fixed RAG method, the number of input tokens is identical across models. In our experiments, the average input token count per question is 1.20k under BM25–Utt and 9.14k under BM25–Session Entire. For OpenAI Embedding, the corresponding averages are 1.13k for Utt and 4.78k for Session Entire.
> The average output tokens per question and the average latency per experiment are shown in Tables 1 and 2. The total cost for each setting can be obtained by multiplying these average input/output token counts by the number of questions (i.e., sessions) in the dataset (e.g., 800 sessions for Friends).
>
> Table 1. The average output tokens per question.
> | Type | Model | BM25 | BM25 | OpenAI Embedding | OpenAI Embedding |
> |------|-------|-----:|------------------:|-----------------:|------------------------------:|
> |      |       | Utt  | Session Entire    | Utt              | Session Entire                |
> | API based  | Gemini 2.5 flash | 1.00  | 1.52  | 1.00  | 1.25  |
> | API based  | Gemini 2.0 flash | 1.00  | 1.10  | 1.00  | 1.05  |
> | API based  | GPT-4o-mini      | 10.21 | 7.59  | 9.33  | 8.50  |
> | API based  | GPT-4.1-nano     | 15.84 | 15.70 | 13.61 | 14.70 |
> | Opensource | Llama 3.3 70b    | 9.41  | 11.77 | 9.16  | 11.84 |
> | Opensource | Llama 3.1 8b     | 48.91 | 17.24 | 68.65 | 12.15 |
> | Opensource | Mixtral          | 56.90 | 26.42 | 62.61 | 36.20 |
> | Opensource | Mistral          | 54.21 | 38.32 | 54.62 | 41.92 |
> | Opensource | Qwen 3 32B       | 18.81 | 155.94| 15.59 | 30.95 |
> | Opensource | Qwen 3 8B        | 11.56 | 39.61 | 10.46 | 20.51 |
> | Opensource | Phi-4-mini 3.8b  | 32.99 | 69.44 | 34.88 | 46.80 |
> | Opensource | Phi-4 14b        | 87.15 | 95.54 | 93.42 | 92.35 |
>
>
> Table 2. The average latency per experiment (hours).
> | Type | Model | BM25 | BM25 | OpenAI Embedding | OpenAI Embedding |
> |------|-------|-----:|------------------:|-----------------:|------------------------------:|
> |      |       | Utt  | Session Entire    | Utt              | Session Entire                |
> | API based  | Gemini 2.5 flash  | 6.71 | 5.52 | 2.60 | 2.72 |
> | API based  | Gemini 2.0 flash  | 6.65 | 5.46 | 2.55 | 2.39 |
> | API based  | GPT-4o-mini       | 6.78 | 5.53 | 2.63 | 2.39 |
> | API based  | GPT-4.1-nano      | 6.76 | 5.55 | 2.61 | 2.37 |
> | Opensource | Llama 3.3 70b     | 7.40 | 8.59 | 3.54 | 4.65 |
> | Opensource | Llama 3.1 8b      | 7.32 | 6.00 | 3.82 | 3.02 |
> | Opensource | Mixtral           | 7.91 | 6.72 | 3.73 | 3.32 |
> | Opensource | Mistral           | 7.19 | 6.14 | 3.13 | 3.02 |
> | Opensource | Qwen 3 32B        | 7.32 | 12.07 | 4.89 | 16.06 |
> | Opensource | Qwen 3 8B         | 6.84 | 7.56 | 2.82 | 3.02 |
> | Opensource | Phi-4-mini 3.8b   | 6.93 | 6.18 | 2.89 | 2.84 |
> | Opensource | Phi-4 14b         | 7.73 | 7.10 | 3.58 | 3.60 |

---

### Meta-Review · Area_Chair_B6ai · 2026-01-04

**Summary:**

## Summary

The paper proposes DialSim, a simulator designed to evaluate the long-term conversation abilities of LLMs using TV series scripts. To support it, the paper introduce a new dataset, which contains dialogs from TV shows like Friends, The Big Bang Theory, and The Office. For every chat, there are different types of questions, some of which are straightforward, some needs memory and reasoning, and some that cannot be answered from the chat context. Experiments show that SOTA LLMs struggle to score high on the benchmark, especially in multi-hop reasoning and long-span dependencies.



## Overall Score

P37e: 6 (no response)

yeJG:  2 (no response)

Kgeb: 6 (no response)

CSQc: 2 (keep the score)

## Concerns

* Limited scalability and generalization (P37e, CSQc)
* Concerns about name swapping (P37e, Kgeb)
* More details about the evaluation (P37e, yeJG, Kgeb, CSQc)
* Limited baselines (yeJG)
* Limited contribution and novelty (CSQc)
* Deeper analysis of the experiments (CSQc, Kgeb)

## Conclusion:

Overall, during the review phase, two reviewers (yeJG and CSQc) weren't fully satisfied with the experimental design and the novelty of the paper. They raised concerns about the limited scalability, weak baselines and potential limitations of name swapping. Reviewer P37e and Kgeb, on the other hand, leaned toward acceptance. During the discussion, the authors did a solid job addressing the reviewers' questions and concerns, but most of the reviewers didn't follow up afterward.

This paper has potential to be stronger with some improvements to the experimental design and deeper analysis of the experiment results.

**Reviewer Concerns:**

Refer to Summary

**Reviewer Scores:**

Refer to Summary

---

### Decision · Program_Chairs · 2026-01-26

Reject